# DECIMER.ai: an open platform for automated optical chemical structure identification, segmentation and recognition in scientific publications

Kohulan Rajan [1], Henning Otto Brinkhaus [1], M. Isabel Agea [2], Achim Zielesny [3] & Christoph Steinbeck [1] ✉

The number of publications describing chemical structures has increased steadily over the last decades. However, the majority of published chemical information is currently not available in machine-readable form in public databases. It remains a challenge to automate the process of information extraction in a way that requires less manual intervention - especially the mining of chemical structure depictions. As an open-source platform that leverages recent advancements in deep learning, computer vision, and natural language processing, *DECIMER.ai* (Deep lEarning for Chemical IMagE Recognition) strives to automatically segment, classify, and translate chemical structure depictions from the printed literature. The segmentation and classification tools are the only openly available packages of their kind, and the optical chemical structure recognition (OCSR) core application yields outstanding performance on all benchmark datasets. The source code, the trained models and the datasets developed in this work have been published under permissive licences. An instance of the *DECIMER* web application is available at https://decimer.ai.

The availability of chemical information in structured data formats and open databases benefits not only researchers in chemistry itself but also scientific fields using chemical information such as medicine, pharmacy, material science, molecular biology and many more[1]. Although substantial efforts exist to establish research data management infrastructures[2,3] and open databases and repositories[4–7], most chemical information is still exclusively published in human-readable text and image formats in the literature. The manual extraction of information from the chemical literature is a time-consuming and error-prone process[8] that can only yield the large amounts of data needed for deep-learning applications,

for example, when considerable amounts of human resources are employed.

The translation of images containing chemical structure depictions into machine-readable representations is referred to as optical chemical structure recognition (OCSR). In the last three decades, there has been continuous development in OCSR tools[9,10], most of them being proprietary algorithms[11] and rule-based tools[12–14]. In general, rule-based tools work better with clean images, whereas slight distortions may hinder their performance[15]. In recent years, deep-learning-based OCSR tools have been developed[16–18] in conjunction with remarkable advancements in computer vision and natural language processing[19,20].

[1]Institute for Inorganic and Analytical Chemistry, Friedrich Schiller University Jena, Lessingstr. 8, 07743 Jena, Germany. [2]Department of Informatics and Chemistry, Faculty of Chemical Technology, University of Chemistry and Technology Prague, Technicka 5, 166 28 Prague, Czech Republic. [3]Institute for Bioinformatics and Chemoinformatics, Westphalian University of Applied Sciences, August-Schmidt-Ring 10, 45665 Recklinghausen, Germany. ✉e-mail: christoph.steinbeck@uni-jena.de

While several publications have claimed to have developed tools that are capable of recognising chemical depictions with high accuracy, most of these tools are either proprietary or entirely unavailable[16,21–23]. Among the few open-source OCSR software solutions[15,24], there is no system that combines chemical structure image segmentation, classification, and translation within a comprehensive workflow.

Here, we present DECIMER.ai, an open-source platform for the identification, segmentation and recognition of chemical structure depictions in the scientific literature that seeks to address this shortcoming. The system combines DECIMER Segmentation, a toolkit based on Mask R-CNN[25] for the detection and segmentation of chemical structures in the scientific literature[26], DECIMER Image Classifier for the identification of images containing a chemical structure, and DECIMER Image Transformer as an OCSR engine, which converts a chemical structure depiction into a machine-readable format. DECIMER algorithms do not inherit any hand-picked rules but instead rely solely on the training data to predict accurate results without making any further hard-coded assumptions.

All components are openly available on GitHub and can be used separately as Python packages or in the user interface of our browser application. The web application is hosted at https://decimer.ai. As all the source code has been published under a permissive licence along with the documentation, users can easily modify and redistribute it or integrate it into their own applications. The Python packages are all hosted on PyPI and are designed to be installable and usable with few lines of code. The DECIMER.ai web application can easily be deployed and scaled. As DECIMER is trained on publicly available data and is made available to the public in the form of a ready-to-use open-source tool, we believe that the system will significantly reduce the workload and produce high-quality data for the research community and those who are developing and curating chemical databases.

## Results

DECIMER Image Classifier and DECIMER Image Transformer have been developed and combined with DECIMER Segmentation[26] to achieve a comprehensive workflow for the automated extraction and interpretation of chemical structures in the scientific literature (Fig. 1). The complete workflow combining all these components is available as a web application with a user interface[27].

DECIMER Image Transformer yields the highest percentage of correct predictions as well as the highest average molecular (Tanimoto) similarities out of all tested tools in our benchmarks (see below). For chemical structure depictions, DECIMER Image Classifier is the first openly available classification system and DECIMER Segmentation[26] is the only openly available segmentation application. The DECIMER web application is the only open-source system that combines these functions in a comprehensive chemical data extraction system.

### DECIMER Image Transformer

The key component of DECIMER.ai is the DECIMER Image Transformer OCSR tool. Due to the usage of diverse chemical structures with diverse depiction features in the training data and an exhaustive image augmentation strategy, the application yields robust results and is capable of interpreting Markush structures as well as common functional groups and superatom abbreviations. A detailed description of the model architecture and the training data is given in the Methods section below.

**In-domain test performance.** The DECIMER Image Transformer model was trained with more than 450 million depictions of chemical structures with an image resolution of 512 × 512 pixels (see dataset pubchem_3 in Supplementary Table 1). The images were generated using the full range of depiction options in the cheminformatics

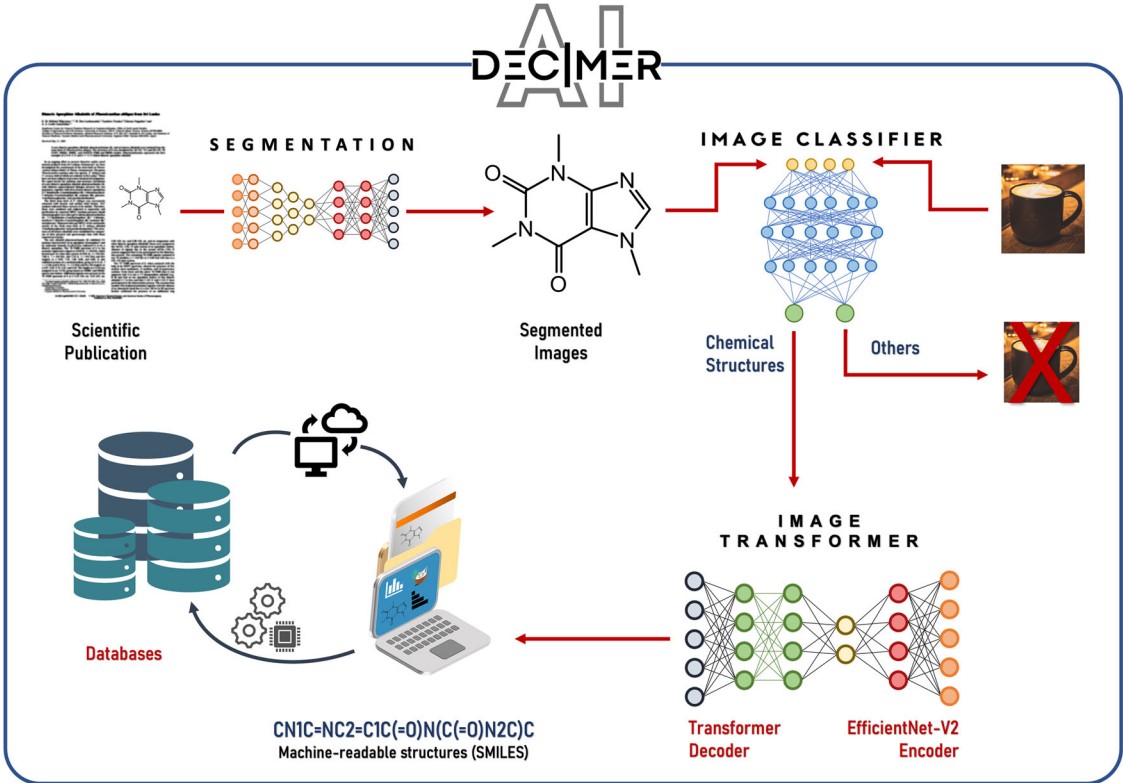

**Fig. 1 | Overview of the integrated DECIMER workflow: detection, segmentation and interpretation of chemical structure depictions in the scientific literature.** A scientific publication is converted into high-resolution PNG images, the Segmentation tool detects and segments chemical structure depictions from the converted images, the Image Classifier checks if the segmented image contains a chemical structure depiction, and finally a machine-readable structure (SMILES) is created from the chemical structure depiction using Image Transformer.

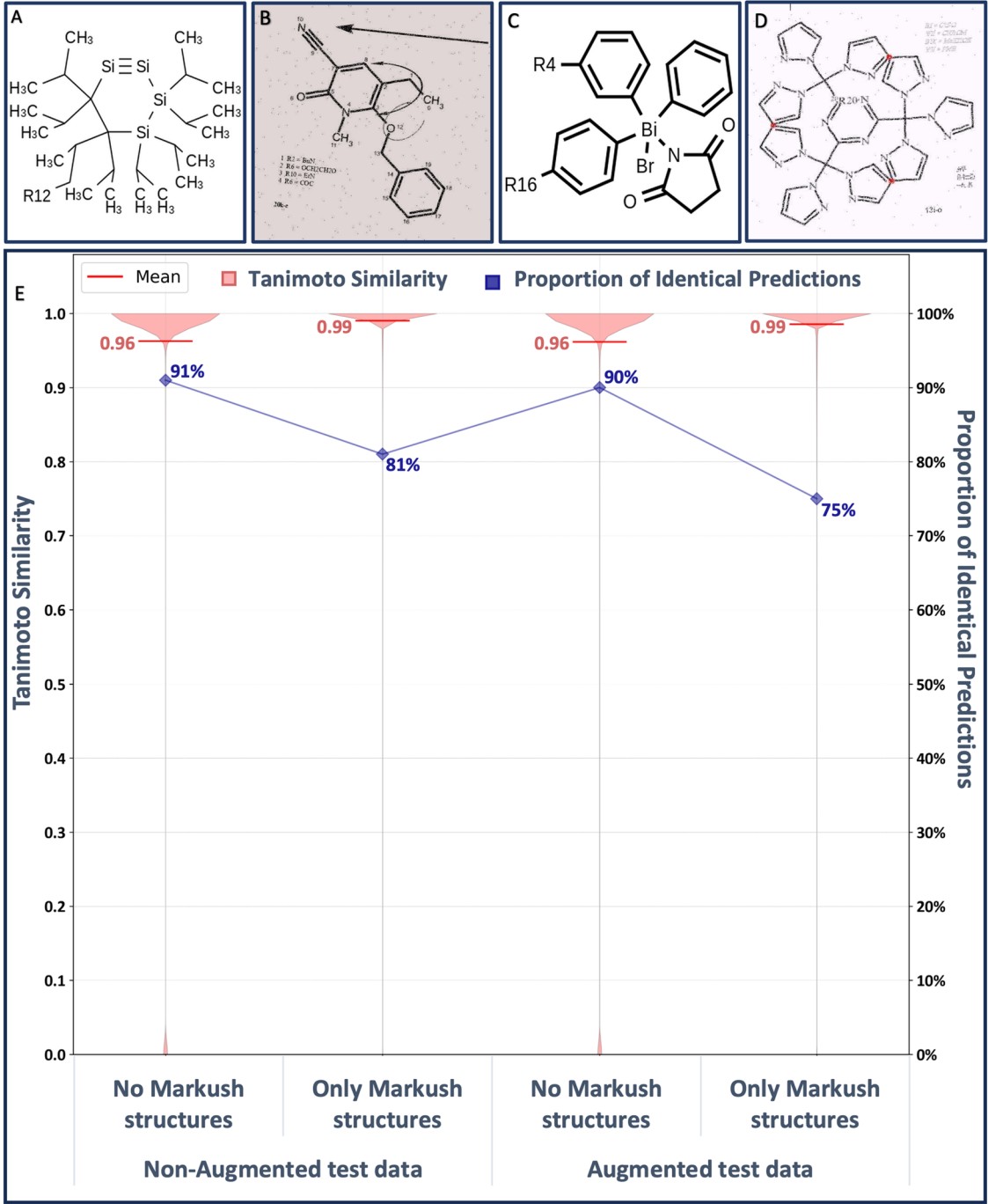

**Fig. 2 | Representation of types of images in the training and the test datasets and in-domain test results. A** Image without augmentations, **B** Image with augmentations, **C** Non-augmented depiction of a Markush structure, **D** Augmented depiction of a Markush structure and **E** In-domain test results: The training dataset includes depictions of Markush structures and a variety of image augmentations (dataset pubchem_3 in Supplementary Table 1). In the test datasets, these features were separately evaluated to assess their influence on performance. All in-domain test results are also presented in Supplementary Table 2.

toolkits Chemistry Development Kit (CDK)[28], RDKit[29], Indigo[30] and the Python-based Informatics Kit for Chemical Units (PIKAChU)[31]. A detailed description of the dataset creation can be found in the Methods section below.

The trained model was tested on four different in-domain datasets containing 250,000 images each. These test datasets were generated similarly to the training datasets but contained no molecules from the training data. In the test datasets, molecules with or without Markush structures or augmentations were included (Fig. 2A).

For performance evaluation, two different measures were used: Predictions identical to the correct molecule were considered to be the optimal result, of course. But predictions resembling the correct molecule closely are also very useful for the curation of chemical data. A human curator, for example, who is presented with the bitmap image and an already very similar machine translation only needs to perform a small correction with a chemical structure editor as opposed to re-drawing the whole molecule. To evaluate the similarity of molecular structures, the Tanimoto similarity[32] or Jacard-Index[33] is used, which encodes the presence or absence of structural features of chemical compounds in a bit vector (where PubChem fingerprints were used in particular) and expresses the similarity between two-bit vectors (or two chemical structures,

**Table 1 | Benchmark results for datasets without added distortions—performance of each model/tool on each dataset**

| Benchmark results for datasets without added distortions. | | | | | | | | | | | | | | | | | | |
|---|---|---|---|---|---|---|---|---|---|---|---|---|---|---|---|---|---|---|
| | JPO | | CLEF | | USPTO | | UOB | | USPTO Big | | Indigo | | Img2Mol Test | | DECIMER-Hand drawn | | DECIMER-Test non-augmented | |
| | $P_i$ | $T$ | $P_i$ | $T$ | $P_i$ | $T$ | $P_i$ | $T$ | $P_i$ | $T$ | $P_i$ | $T$ | $P_i$ | $T$ | $P_i$ | $T$ | $P_i$ | $T$ |
| OSRA | 56% | 0.78 | **85%** | 0.88 | **88%** | 0.96 | 78% | 0.95 | 0.01% | 0.17 | 2% | 0.29 | 2% | 0.14 | 1% | 0.17 | 8% | 0.33 |
| MolVec | **66%** | 0.89 | 83% | 0.89 | **88%** | 0.97 | 80% | 0.96 | 1% | 0.35 | 2% | 0.27 | 2% | 0.29 | 1% | 0.23 | 5% | 0.33 |
| Imago | 40% | 0.68 | 59% | 0.85 | 87% | 0.96 | 58% | 0.87 | 0% | 0.10 | 0.04% | 0.08 | 0.02% | 0.11 | 3% | 0.22 | 2% | 0.19 |
| Img2Mol | 15% | 0.70 | 16% | 0.81 | 24% | 0.85 | 68% | 0.94 | 16% | 0.78 | 22% | 0.59 | **85%** | **0.97** | 5% | 0.52 | 16% | 0.78 |
| SwinOCSR | 13% | 0.75 | 29% | 0.81 | 27% | 0.88 | 45% | 0.97 | 0.23% | 0.68 | 0.20% | 0.48 | 4% | 0.53 | 5% | 0.64 | 6% | 0.54 |
| MolScribe | 50% | 0.93 | 75% | 0.89 | 79% | **0.99** | 87% | **0.99** | **79%** | 0.95 | 38% | 0.65 | 51% | 0.93 | 8% | 0.59 | 44% | 0.85 |
| DECIMER | 64% | **0.93** | 72% | **0.96** | 61% | 0.97 | 88% | 0.98 | 63% | **0.97** | **60%** | **0.98** | 55% | 0.93 | 27% | 0.69 | **91%** | **0.99** |
| DECIMER Fine Tuned | - | - | - | - | - | - | - | - | - | - | - | - | - | - | 60% | 0.89 | - | - |

The performance is described as the proportion of occurrences of identical predictions $P_i$ and the average Tanimoto similarity **$T$**.
The best result for each metric on each dataset is marked in bold.

**Table 2 | Benchmark results for datasets with added distortions, such as mild shearing and rotation—performance of each model/tool on each dataset**

| Benchmark results for datasets with distortions | | | | | | | | | | | | | | |
|---|---|---|---|---|---|---|---|---|---|---|---|---|---|---|
| | JPO (dist) | | CLEF (dist) | | USPTO (dist) | | UOB (dist) | | USPTO_big (dist) | | Indigo (dist) | | DECIMER-Test augmented | |
| | $P_i$ | $T$ | $P_i$ | $T$ | $P_i$ | $T$ | $P_i$ | $T$ | $P_i$ | $T$ | $P_i$ | $T$ | $P_i$ | $T$ |
| OSRA | 38% | 0.70 | 19% | 0.66 | 7% | 0.60 | 61% | 0.90 | 0.01% | 0.13 | 0.42% | 0.16 | 2% | 0.15 |
| MolVec | 41% | 0.80 | 21% | 0.66 | 26% | 0.71 | 63% | 0.92 | 0.02% | 0.14 | 0.48% | 0.07 | 1% | 0.12 |
| Imago | 23% | 0.47 | 33% | 0.65 | 51% | 0.81 | 34% | 0.64 | 0% | 0.08 | 0.01% | 0.20 | 0.15% | 0.10 |
| Img2Mol | 15% | 0.67 | 15% | 0.80 | 21% | 0.83 | 70% | 0.94 | 1% | 0.56 | 15% | 0.54 | 1% | 0.60 |
| SwinOCSR | 7% | 0.71 | 21% | 0.81 | 23% | 0.87 | 6% | 0.95 | 0% | 0.38 | 0.01% | 0.38 | 0.18% | 0.36 |
| MolScribe | 52% | **0.93** | **73%** | 0.89 | **75%** | **0.99** | 86% | **0.99** | 78% | 0.95 | 34% | 0.64 | 9% | 0.53 |
| DECIMER | **62%** | **0.93** | 72% | **0.96** | 61% | 0.96 | **86%** | 0.98 | 57% | **0.96** | **51%** | **0.97** | **90%** | **0.99** |

The performance is described as the proportion of occurrences of identical predictions $P_i$ and the average Tanimoto similarity **$T$**.
The best result for each metric on each dataset is marked in bold.

respectively) as a number between 0.0 (most dissimilar) and 1.0 (most similar).

In all test results, DECIMER Image Transformer consistently produces an average Tanimoto similarity of greater than 0.95 (Fig. 2). Opposed to the steadily high Tanimoto similarity, there are clear differences regarding the number of perfect predictions. The proportion of perfectly predicted molecules decreases with an increased level of complexity and noise in the structure depictions as well as a lower image resolution.

There are two obvious trends: (1) The addition of image augmentations leads to a lower proportion of perfectly recognised structures. (2) The proportion of perfectly recognised molecules is lower when processing test datasets that exclusively contain Markush structures. These results are not surprising since the R-group indices (as in ‘R$_1$’) and other labels can be difficult to recognise, especially when the image resolution is low or when additional noise is introduced. Nevertheless, the constantly high Tanimoto similarities indicate that the predicted molecules are very similar to the depicted ones, even when the predictions are not perfect.

Since PubChem fingerprints cannot describe the R-group variables in Markush structures, the derived Tanimoto similarities only describe the similarities of the molecular structures, but cannot be used to evaluate whether the R-group labels have been correctly interpreted. Therefore, the BLEU score[34] was determined as a token-based string similarity metric (see Supplementary Table 2). The obtained average BLEU score of 0.94 across all test results with

Markush structures also indicates a high similarity between the predicted and the true string representations.

**OCSR tools benchmark.** To assess the performance of the DECIMER Image Transformer model in comparison with other openly available tools (OSRA[12], MolVec[14], Imago[13], Img2Mol[15], MolScribe[35], SwinOCSR[24], see Tables 1 and 2), a row of benchmark datasets from a variety of sources was applied (a complete list with additional information about the benchmark datasets and individual tool performance is provided in the Methods section and the Supplementary information). Following the remark of Clévert et al.[15], that the parameters of the rule-based systems OSRA, MolVec and Imago are overfitted to the available benchmark datasets, mild image distortions (i.e., rotations in the range between −5° and + 5° and mild shearing) were applied to all datasets (see Fig. 3B/3D in contrast to Fig. 3A/3C for datasets without these distortions). The detailed performance metrics for every tool on every benchmark dataset are presented in Tables 1–4.

DECIMER Image Transformer achieves competitive results on most benchmark datasets compared to the other open OCSR tools (Fig. 3), showing no performance degradation due to slight image distortions while confirming the lack of distortion robustness of the rule-based systems. In addition, the rule-based systems fail to correctly recognise the structure depictions with a low image resolution (see USPTO_big and Indigo in Table 2). The SwinOCSR model does not achieve any outstanding results in our benchmarks−but it needs to be mentioned that its developers stated that their model does not

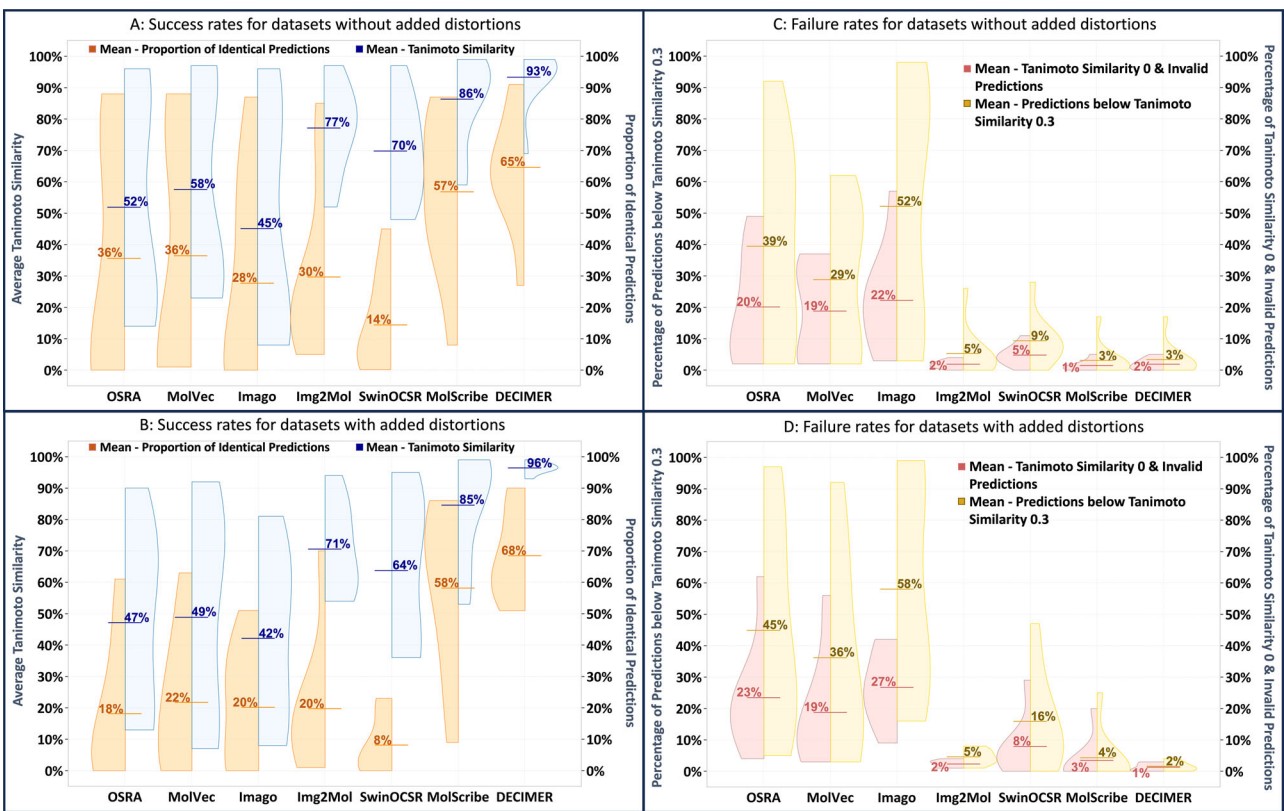

**Fig. 3 | Average performance of the open OCSR tools on all benchmark datasets.** The success rates are described by the proportion of perfect predictions and the average Tanimoto similarities, whereas the failure rates are measured as the percentage of predictions with zero Tanimoto similarity plus invalid predictions (catastrophic) and the percentage of predictions with a low Tanimoto similarity value less than or equal to 0.3 (severe). **A** Success rate for datasets without added distortions. **B** Success rates for datasets with added distortions. **C** Failure rates for datasets without added distortions. **D** Failure rates for datasets with added distortions.

perform well on real-world data[24], which is likely due to a lack of diversity in their training data. Appropriate assessment of failure rates is of particular importance for machine learning applications (Fig. 3C/3D): in line with Img2Mol and MolScribe, DECIMER Image Transformer exhibits extremely low rates of severe and catastrophic failures.

The DECIMER Image Transformer model has never been trained on hand-written chemical structure depictions. However, for a benchmark dataset that only consists of hand-drawn chemical structures (DECIMER Hand-drawn image dataset outlined in the Methods section), it recognises 27% of the structures perfectly and achieves an average Tanimoto similarity of 0.69, whereas all alternative open tools perform worse (see Table 1). Moreover, when the model is fine-tuned with a training dataset of images with augmentations that make them appear hand-drawn-like (see Fig. 4 and Supplementary Table 1), the proportion of perfect predictions grows significantly to 60% (i.e., an increase of 33%), corresponding to a remarkable average Tanimoto similarity increase of plus 0.2 to 0.89.

## DECIMER Image Classifier
DECIMER Image Classifier is a deep-learning-based architecture for the identification of images that contain a depiction of a chemical structure. It has been trained, tested and validated using a balanced dataset of images with and without chemical structure depictions (creation and curation of this dataset are outlined in detail in the Methods section).

In addition, DECIMER Image Classifier has been tested on four external datasets, three publicly available datasets

1. a dataset only containing chemical structure depictions (ChEBI),
2. a dataset without any chemical structure depictions (EM_Images),

3. a public dataset of images extracted from a diverse set of publications (PubLayNet),

and a manually curated set of images extracted from articles of the Journal of Natural Products (JNP):
4. a real-world dataset using 1000 publications (JNP_real_world).

DECIMER Image Classifier predicts a value between 0 and 1, where an optimally determined threshold value is used for the binary decision purpose. The system achieved a 0.99 score on the in-domain test set on every performance metric calculated (Area Under Curve, Matthews Correlation Coefficient, accuracy, specificity, and sensitivity, see Methods section below). It correctly classified 99% of the images with chemical structure depictions and almost 100% of images without chemical structure depictions. On the four out-of-domain test sets, the proportion of true classifications was 97% (ChEBI), 100% (EM_Images), 99% (PubLayNet) and 94% (JNP_real_world).

## DECIMER.ai
DECIMER.ai is a web application that combines the previously described components in an automated, comprehensive workflow for the extraction of chemical structures from the scientific literature. When a user uploads a PDF document or a single image file, DECIMER Segmentation is used to cut chemical structure depictions. The segmented chemical structure depictions are then processed by DECIMER Image Classifier and DECIMER Image Transformer to obtain machine-readable SMILES string representations of the resolved chemical structures.

**Table 3 | Benchmark results for datasets without added distortions—Catastrophic and severe failure rates of each model/tool on each dataset**

Benchmark results for datasets without added distortions.

| | JPO | | CLEF | | USPTO | | UOB | | USPTO Big | | Indigo | | Img2Mol Test | | DECIMER-Hand drawn | | DECIMER-Test non-augmented | |
|---|---|---|---|---|---|---|---|---|---|---|---|---|---|---|---|---|---|---|
| | $T_E$ | $T_{<=0.3}$ | $T_E$ | $T_{<=0.3}$ | $T_E$ | $T_{<=0.3}$ | $T_E$ | $T_{<=0.3}$ | $T_E$ | $T_{<=0.3}$ | $T_E$ | $T_{<=0.3}$ | $T_E$ | $T_{<=0.3}$ | $T_E$ | $T_{<=0.3}$ | $T_E$ | $T_{<=0.3}$ |
| OSRA | 14% | 19% | 4% | 4% | 2% | 2% | 2% | 2% | 8% | 92% | 25% | 42% | 34% | 63% | 49% | 73% | 43% | 58% |
| MolVec | 6% | 8% | 3% | 3% | 2% | 2% | 2% | 2% | 21% | 45% | 28% | 35% | 36% | 55% | 34% | 62% | 37% | 47% |
| Imago | 23% | 25% | 7% | 7% | 3% | 3% | 6% | 7% | 19% | 98% | 23% | 92% | 27% | 91% | 57% | 67% | 35% | 79% |
| Img2Mol | 2% | 7% | 3% | 3% | 3% | 3% | 1% | 1% | 1% | 2% | 1% | 2% | **0.29%** | **0.32%** | **2%** | 26% | 4% | 4% |
| SwinOCSR | 6% | 9% | 5% | 6% | 2% | 3% | 0.21% | 0.33% | 3% | 6% | 5% | 8% | 8% | 12% | 3% | **12%** | 11% | 28% |
| MolScribe | **1%** | **2%** | 3% | 3% | **0.37%** | **0.4%** | 0.02% | 0.02% | **0.22%** | **0.23%** | **1%** | **1%** | 1% | 2% | 5% | 17% | **2%** | **3%** |
| DECIMER | 3% | 3% | **2%** | **2%** | 1% | 1% | **0%** | **0%** | 0.25% | 0.45% | **0.20%** | **0.21%** | 2% | 3% | 5% | 17% | 4% | 4% |

$T_E$: Percentage of predictions with Tanimoto similarity values of zero and invalid predictions (catastrophic failure). $T_{<=0.3}$: The percentage of predictions with Tanimoto similarity less than or equal to 0.3 (severe failure).
The best result for each metric on each dataset is marked in bold.

**Table 4 | Benchmark results for datasets with added distortions, such as mild shearing and rotation—Catastrophic and severe failure rates of each model/tool on each dataset**

B. Benchmark results for datasets with distortions

| | JPO (dist) | | CLEF (dist) | | USPTO (dist) | | UOB (dist) | | USPTO_big (dist) | | Indigo (dist) | | DECIMER-Test augmented | |
|---|---|---|---|---|---|---|---|---|---|---|---|---|---|---|
| | $T_E$ | $T_{<=0.3}$ | $T_E$ | $T_{<=0.3}$ | $T_E$ | $T_{<=0.3}$ | $T_E$ | $T_{<=0.3}$ | $T_E$ | $T_{<=0.3}$ | $T_E$ | $T_{<=0.3}$ | $T_E$ | $T_{<=0.3}$ |
| OSRA | 18% | 23% | 19% | 20% | 25% | 26% | 4% | 5% | 11% | 97% | 25% | 62% | 62% | 81% |
| MolVec | 10% | 12% | 12% | 13% | 15% | 16% | 3% | 3% | 5% | 92% | 30% | 50% | 56% | 67% |
| Imago | 42% | 46% | 28% | 29% | 16% | 16% | 27% | 29% | 9% | 99% | 23% | 95% | 42% | 92% |
| Img2Mol | 3% | 7% | 3% | 4% | 3% | 3% | 1% | 1% | 1% | 6% | 1% | 3% | 4% | 8% |
| SwinOCSR | 5% | 10% | 5% | 6% | 2% | 3% | 0.14% | 0.23% | 7% | 28% | 7% | 17% | 29% | 47% |
| MolScribe | **0.44%** | **1%** | 3% | 3% | **0.39%** | **0.43%** | **0%** | **0%** | **0.23%** | **0.27%** | 1% | 1% | 3% | 25% |
| DECIMER | 3% | 4% | **2%** | **2%** | 1% | 1% | **0%** | **0%** | 0.39% | 0.74% | **0.16%** | **0.19%** | **3%** | **3%** |

$T_E$: Percentage of predictions with Tanimoto similarity values of zero and invalid predictions (catastrophic failure). $T_{<=0.3}$: The percentage of predictions with Tanimoto similarity less than or equal to 0.3 (severe failure).
The best result for each metric on each dataset is marked in bold.

These resolved SMILES encoded structures are then automatically loaded in the embedded molecular editor Ketcher[36] (see Fig. 5). The molecular editor enables the manual inspection and editing of the resolved chemical structures. In addition to the segmented structure depictions, the resolved structures can be downloaded in the common MOL file format.

## Discussion

DECIMER Image Transformer as the DECIMER core component achieves highly accurate results on the in-domain test data. The system performs better on non-augmented test images since augmented images contain a wide range of additional non-structural elements and noise that have to be ignored in order to correctly translate a chemical structure depiction. This effect is diminished when images of a higher resolution are processed because a low-resolution image may already be comparably blurry and may turn unrecognisable when additional augmentations are applied. The DECIMER Image Transformer model produces predictions that are highly similar to the original molecules with an average Tanimoto similarity over 0.95. The translation of depictions of Markush structures yields similar results, although the proportion of perfectly predicted structures is considerably lowered.

This may be traced to the relevance of the small subscript indices of the R-groups (as in '$R_1$'). It could be shown that especially in images with a lower resolution of $299 \times 299$ pixels, these small digits may become unrecognisable, whereas corresponding images with a resolution of $512 \times 512$ pixels could be processed with a significant increase in the number of perfectly recognised Markush structures (see Supplementary Table 2 and Supplementary Fig. 1). Moreover, the BLEU scores, which are consistently above 0.9 (see Supplementary Table 2), confirm that the Markush structure predictions are very similar to the original SMILES strings.

In comparison with alternative open OCSR tools, DECIMER Image Transformer performs with high accuracy. Apart from Img2Mol's performance on its in-domain test data, MolScribe's performance on USPTO data (which may be part of the system's training data[35]) and the performance of MolVec on the non-distorted JPO, CLEF and USPTO datasets, DECIMER Image Transformer performs outstandingly well on all benchmark datasets without any significant differences between non-distorted and distorted images. It is particularly striking that the system's severe and catastrophic failure rates are very low. The system also achieves a comparative peak performance when benchmarked against the DECIMER Hand-drawn image dataset, which is especially

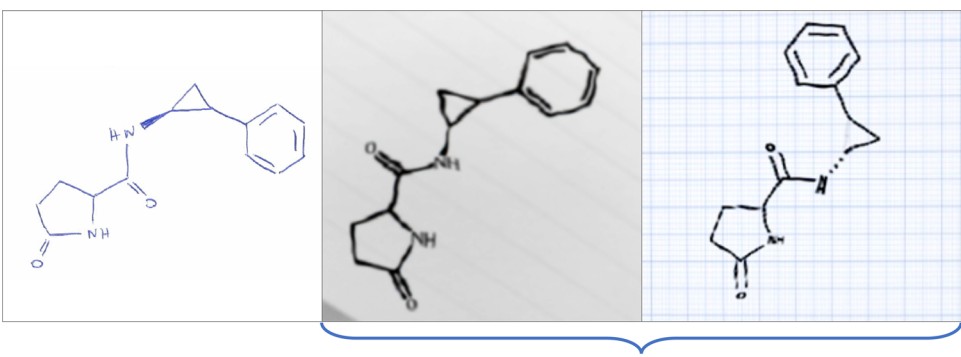

Hand drawn image    Hand-drawn-like synthetic images

**Fig. 4 | Comparison of a hand-drawn molecule and synthetic hand-drawn-like images.** A hand-drawn molecule representation from the *DECIMER Hand-drawn* image dataset[67] (PubChem ID: 31743 [https://pubchem.ncbi.nlm.nih.gov/compound/31743], left) and corresponding synthetic hand-drawn-like images created with RanDepict[37] (middle, right).

**Resolved SMILES representation**
CN1C2=C(C(=[X])N(C1=O)[R2])N(C=N2)[R1] – Search for this structure on PubChem

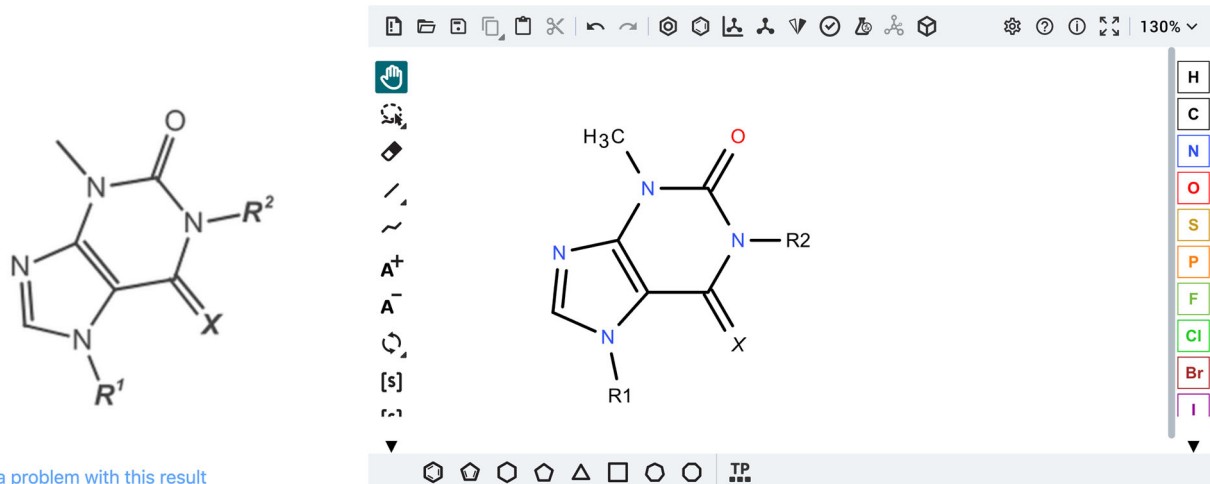

Report a problem with this result

**Fig. 5 | Example image of a Markush structure (on the left) that has been loaded into the DECIMER web application.** The SMILES string representation of the molecule is generated (upper left) and depicted in the embedded Ketcher molecular editor window (on the right).

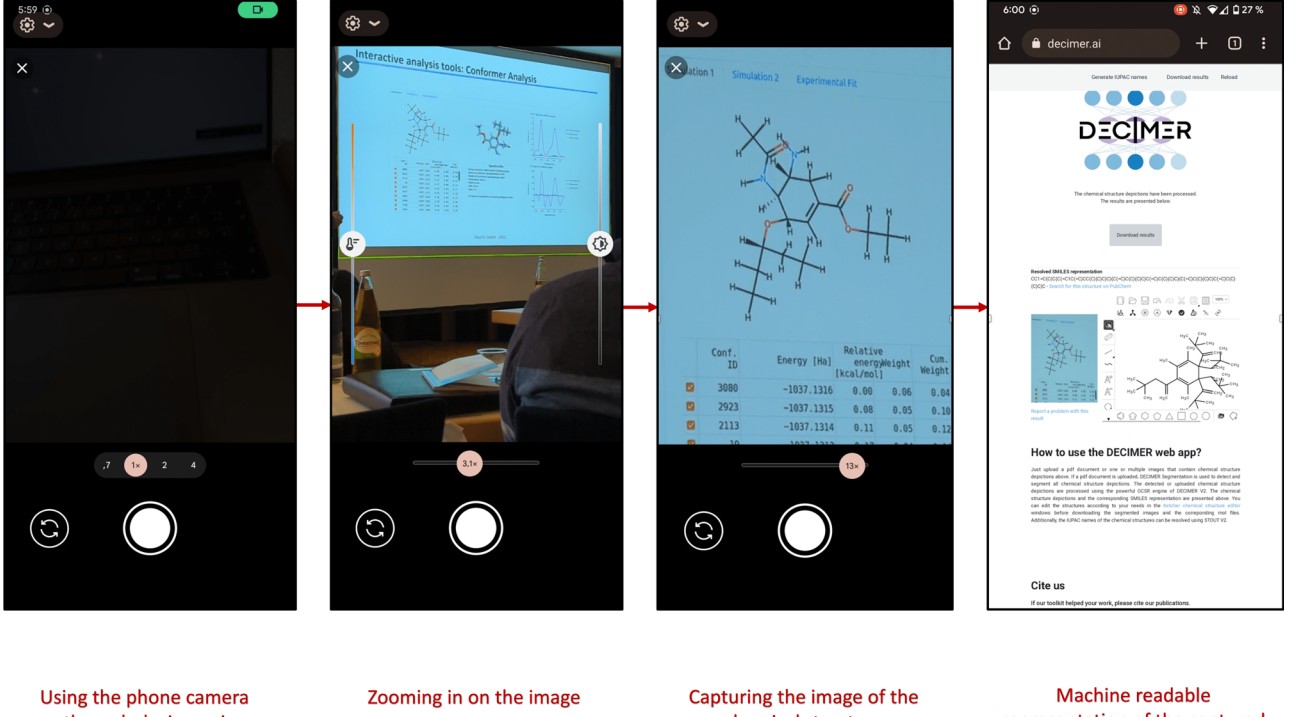

| Using the phone camera through decimer.ai | Zooming in on the image | Capturing the image of the chemical structure | Machine readable representation of the captured image |

**Fig. 6 | DECIMER.ai being used via a smartphone at the 17th German Conference on Cheminformatics.** The deciphered structure can be searched in PubChem, the largest openly available chemical database, right away.

interesting since there has not been a single hand-drawn structure in the training data. Thus, this model may be applied to extract chemical structures from hand-drawn images in the future. This may become particularly relevant for translating chemical publications from 50 years ago since a lot of the chemical structures from that time were hand-drawn using templates. Although the predictions might not be perfect in all cases, a similar prediction considerably reduces the amount of manual work when mining chemical structures from printed literature. The outlined success rates of DECIMER Image Transformer demonstrate not only robust performance but also superior generalisation capabilities due to a training data diversification strategy with highly diverse structure depictions generated by our OCSR training data generation tool RanDepict[37]: It ensures that the full diversity of depiction features is properly represented that CDK[28], RDKit[29], Indigo[30], and PIKAChU[31] have to offer.

DECIMER Image Classifier is capable of achieving high-performance metrics and is capable of working effectively on a wide range of datasets. In the ChEBI dataset, performance was slightly reduced due to the presence of images of isolated ions that were recognised as non-chemical images. None of the electron microscopy images from the EM_Images dataset has been wrongly classified as a chemical structure. Considering that the images found in PubLayNet originated from diverse sets of articles from PubMed Central, the high performance of the DECIMER Image Classifier indicates the robustness of the model. Additionally, the classifier achieved high performance when applied to real-world use cases.

The basic OCSR approach of DECIMER can be described as a direct mapping from the entity (graphical image) to entity (SMILES representations of chemical structure) without any intermediate steps. In creating a large training dataset for the DECIMER Image Transformer, we have attempted to cover a large portion of the chemical space in order to create a robust model that can interpret most types of structure representations found in the literature. The high performance of the DECIMER Image Transformer in the benchmark analysis

confirms that this has been achieved. Interestingly, MolScribe performs exceptionally well on the majority of benchmark datasets, achieving similar accuracy to DECIMER despite being trained on a much smaller dataset (1.68 million images versus over 400 million images). MolScribe employs a different model architecture with pre-defined rules to reconstruct a molecular graph from predicted atoms and bonds with coordinates, in contrast to DECIMER's purely data-driven approach. By training DECIMER Image Transformer from scratch with the small MolScribe training dataset and comparing their predictive performance achieved with the same training data, it becomes clear that MolScribe consistently outperforms DECIMER on the benchmark datasets. The results suggest that MolScribe's model architecture and integrated post-processing workflow contribute to its efficiency, while DECIMER's exclusively data-driven approach exhibits its predictive abilities only on large training datasets. In addition, significant progress has been made in the area of object detection-based OCSR systems[21,38,39]. These systems recognise different structural elements in a given image, which are then used to construct a molecular graph. Hormazabal et al. have shown that object detection-based OCSR systems can achieve good results on common benchmark datasets with relatively small training datasets[39]. To our knowledge, none of these systems are openly available, so we could not include them in our benchmark analysis. In summary, it will be mutually beneficial to pursue all these promising different approaches to the OCSR task in order to obtain an increasingly clear picture of their principal predictive capabilities as well as their different learning efficiencies, although final statements are not yet possible.

The DECIMER.ai web application is the first comprehensive open-source user interface application for the extraction of chemical information from scientific literature. As discussed above, DECIMER Image Transformer translates chemical structure depictions with a high degree of similarity. By embedding it into the DECIMER.ai application, a human curator can immediately assess the predictions and correct them in the molecular editor windows if necessary. For the

segmentation and classification of chemical structure depictions, DECIMER Segmentation and DECIMER Image Classifier are the only open-source applications available.

Since DECIMER.ai can be accessed from a mobile phone or tablet via the web browser, these tools are enabled to recognise chemical structures in the real-world (Fig. 6): By using DECIMER.ai on a mobile device during a conference, images of chemical structures may be captured during a presentation or poster session to identify the molecules presented. With the DECIMER.ai search functionality, users can conduct a direct "single-click" PubChem database search in addition to the structure recognition to access additional chemical information.

There have been closed-source projects like CLiDE[40] or the recently published MolMiner[23] that combine a segmentation step with an OCSR step in their workflow. CLiDE is a fully commercial tool, MolMiner permits limited access to registered users and offers unlimited access to users who wish to obtain an enterprise licence. Since the source code of these applications is not openly available, researchers cannot adapt them according to their needs or integrate them into their applications. As all DECIMER components and the DECIMER.ai web application are open-source projects, continuous further development with significant community-driven improvements can be expected in the future. There have been major advances in the extraction of chemical information from documents. For example, ChemDataExtractor[8,41,42] has been used extensively for the automated generation of chemical databases[43–46]. Perspectively, it would be interesting to integrate such applications in DECIMER.ai to mine chemical information from the text of PDF documents and link it to structural information obtained from OCSR. Although there are many more challenges to overcome to mine all types of chemical information from the literature using a single platform, DECIMER.ai may become a solid open basis for further development.

## Methods

The DECIMER project was developed as a deep-learning-based solution for OCSR tasks. The goal of the DECIMER project is to develop an automated system that detects, segments, and converts images from published literature into computer-readable formats, in this case, the SMILES representation. It is a fully data-driven approach, in which no assumptions are made about the underlying chemical structure. In total, the project is divided into four parts: the segmentation algorithm, the image classifier, the OCSR model, and the web application.

### DECIMER segmentation

Our previously published application DECIMER Segmentation[26] was re-used in this work to create a complete extraction workflow. It uses an open implementation of the Mask R-CNN architecture[25] in combination with custom processing steps to segment chemical structure depictions from pages in the scientific literature. Since the original publication, we have refactored the complete codebase, added unit tests and wrapped it up in a Python package that can be installed easily from PyPI[47], but all underlying models and algorithms remain unchanged. The DECIMER Segmentation model was trained on manually annotated data using TensorFlow 2.3.0, but it has been updated to work with TensorFlow 2.10.0 in accordance with the other DECIMER components. The source code and the model are available on GitHub[48] and Zenodo[49]. For further information about DECIMER Segmentation, we would like to refer to the original publication[26].

### DECIMER Image Transformer

**Selection of molecules.** The DECIMER Image Transformer model was trained on data based on molecules obtained from PubChem[50]. The entire molecules of PubChem were downloaded in SMILES format directly from the PubChem FTP site[50]. To reduce the imbalance of data, all molecules with a molecular weight of more than 1500 Dalton were filtered out. All explicit hydrogen atoms were removed and stereochemistry was retained. SMILES strings with more than 152 tokens (see Tokenization) were filtered out due to their underrepresentation in the data (3263 molecules). As a result, 108,541,884 molecules were selected in total. A diverse set of 250,000 molecules was selected to use as a test dataset from the whole dataset using the MaxMin[51] algorithm included in chemfp[52]. Another million molecules were selected randomly and used for validation during the development, and the remainder was used as a training dataset (pubchem_1 see Supplementary Table 1).

Additionally, a second dataset with Markush structures was generated. Due to the unavailability of large datasets of SMILES that represent Markush structures, they were artificially generated based on 20 million SMILES strings which were diversely picked from PubChem[50] using the chemfp[52] implementation of the MaxMin[51] algorithm. To generate SMILES representing Markush structures, the following steps were followed:

1. Read input SMILES using the CDK[28].
2. Add explicit hydrogen atoms and return absolute SMILES.
3. Pseudo-randomly replace 1-3 carbon-'C' or hydrogen-'H' with the rest group variables. Rest group variables are defined as the characters 'R', 'X', and 'Z' with or without an index number between 0 and 20.
4. Read modified SMILES using the CDK.
5. Remove explicit hydrogen atoms and return absolute SMILES.

For example, the input SMILES string 'CCC' is converted to the absolute SMILES string 'C([H])([H])([H])C([H])([H])C([H])([H])[H]'. Subsequently, the pseudo-random insertion of an R-group variable takes place and yields 'C([H])([H])([H])C([H])([H])C([H])([R])[H]'. After re-reading the modified SMILES string and removing the explicit hydrogen atoms, the CDK returns 'CC(C)[R]'. The functionality of generating random Markush structures based on given SMILES strings has been integrated into our open-source OCSR training data generation tool RanDepict[37] for this purpose.

By adding the newly generated SMILES with Markush structures to the SMILES strings from pubchem_1 and applying the same filtering criteria as described above, 126,702,705 molecules were selected. Based on this, a diverse set of 250,000 SMILES representing molecules with Markush structures were selected for testing using the MaxMin[51] algorithm. One million molecules were retained to use for validation during development, and the remainder was used as training data (pubchem_2, pubchem_3 see Supplementary Table 1).

Our previous study on the performance of the molecular string representations DeepSMILES[53], SELFIES[54] and SMILES for OCSR purposes[55] with similar model architectures indicates that the usage of SMILES strings leads to more accurate results although the usage of SELFIES leads to more valid chemical structures in the predictions. Thus, SMILES string representations were used for DECIMER Image Transformer.

**Tokenization.** The SMILES strings in the datasets were split into meaningful tokens using the Keras[56] tokenizer with TensorFlow 2.8.0[57]. The following set of rules was applied where each string is split after,

- every heavy atom: e.g., "C", "Si", "Au"
- every open bracket and closed bracket: "(", ")", "[","]"
- every bond symbol: "=","#"
- every one of the following characters: ".", "-","+","\","/","@","%","*"
- every single-digit number

After the splitting, a "<start>" and an "<end>" token were added at the beginning and the end of the sequence. To match the same maximum length, each tokenized string was padded with "<pad>" tokens. The token "<unk>" is used for unknown elements and acts as a placeholder. R-group indices were replaced according to the procedure

described in the subsection Evaluation of different R-group representations in SMILES.

The following is a list of all tokens found in dataset pubchem_1: <unk>, C, =, (,), O, N, 1, 2, 3, <start>, <end>, @, [,], 4, H, F, S, 5, Cl, /,., 6, −, +, Br, #, \, 7, 8, 9, P, I, Si, B, Na, K, %, Se, Sn, Y, Li, Zr, Fe, Ti, Al, Zn, Pt, Cu, Ir, Mg, Ni, Co, W, Ru, Ca, Ge, V, As, 0, Pd, Cr, Mn, Sb, Ag, Te, Hg, Mo, Hf, Rh, Au, Pb, Ba, Bi, U, Rb, In, Cs, Ga, Re, Cd, Ar, Sr, Os, Ce, La, Gd, Tl, Nb, Nd, Ta, Eu, Pr, Sm, Yb, Sc, Be, Tb, Dy, Er, Th, Lu, Ho, *, Tm, Xe, He, Pa, Kr, Ne, <pad>

The following is a list of all tokens found in dataset pubchem_2: <unk>, C, =, (,), O, N, 1, 2, [,], 3, <start>, <end>, @, 4, H, F, S, Cl, 5, /, !, X, Z, R,., 6, Br, +, −, #, \\, §, $, 7, £, <, ?, ¢, ^, >, €, 8, I, P, 9, Si, B, Na, %, Se, 0, Sn, K, Y, Li, Zr, Fe, Al, Ti, Zn, Pt, Cu, Ir, As, Ni, Mg, Ge, W, Co, Ru, Ca, V, Pd, Te, Cr, Mn, Sb, Hg, Ag, Mo, Pb, Hf, Bi, Au, Rh, Ba, U, In, Rb, Ga, Re, Cs, Cd, Sr, Ar, Tl, Ce, Os, La, Nb, Gd, Ta, Nd, Eu, Pr, Sm, Yb, Sc, Be, Tb, Th, Er, Dy, Lu, Ho, *, Tm, Xe, He, Kr, Pa, Ne, <pad>

**Generation of chemical structure depictions.** The images of chemical structures were depicted as grayscale 2D bitmap images using our open-source toolkit RanDepict[37]. In the chemical literature, various types of chemical structure depictions are represented. This is due to the usage of numerous different software packages or even templates for hand-drawing chemical structures throughout different types of publications. RanDepict attempts to generate datasets in which all features that define different types of depictions are represented in a balanced and controlled manner by pseudo-randomly scrambling all available depiction parameters for every created image. The latest version of RanDepict uses the depiction functionalities of the CDK[28], RDKit[29] and Indigo[30] and PIKAChU[31] to achieve this. The depiction parameters comprise, for example, the orientation of the depicted molecule, the bond width and length, the font type and size used for text labels, the distance between lines and text labels in the structure and the presence of chirality labels. The CDK, RDKit and Indigo offer functionalities to replace substructures with text labels. These functionalities are used to include superatom and functional group labels in the structure depictions. For example, a phenyl group may be represented as a fully drawn substructure or the text label 'Ph'. In both cases, they are mapped to the same SMILES substring. Additionally, a variety of image augmentations like rotation, shearing, random black and white noise, pixelation, the addition of curved arrows in a structure, and the addition of text labels and reaction arrows around the structure are applied (see examples in Fig. 2A). For more details about the generation of diverse sets of chemical structure depictions with various image augmentations, we would like to refer to the publication of the original version of RanDepict[37].

The originally published version of RanDepict (1.0.5) uses the CDK[28], RDKit[29] and Indigo[30] toolkits to generate diverse sets of chemical structure depictions. For the training dataset pubchem_1, this version of RanDepict was used to depict each molecule once without and three times with image augmentations with different pseudo-randomly scrambled depiction parameters for each image.

Since then, we have continued the development of RanDepict and have implemented the option to depict Markush structures. Additionally, the generation of SMILES representations of Markush structures based on any given SMILES string that has been described in the section Selection of molecules was implemented. Furthermore, we contributed to PIKAChU[31], to allow the depiction of Markush structures and implemented its functionalities in RanDepict. Finally, RanDepict 1.0.8 was used to generate the chemical structure depictions in the training dataset pubchem_2, which contains Markush structures where the images were depicted with a size of 299 × 299 pixels. Here once again, one depiction was created without any augmentations, and three depictions were created with augmentations. This version of RanDepict produced some invalid SMILES representations of Markush structures resulting in a reduction of total images. Due to the large

number of depictions (479,500,000 images) and the time and resources spent on their production, we decided to proceed with this dataset.

To evaluate the performance of the model using images with a higher resolution, a third dataset was created by re-depicting the molecules from the pubchem_2 dataset with an image size of 512 × 512 pixels (where originally the images on pubchem_2 dataset were depicted with an image size of only 299 × 299 pixels). Everything else was done following the same procedure as the production of pubchem_2. During the creation of the dataset, not all molecules were completely depicted due to memory issues, resulting in a reduction in the number of images. Again, we decided to use the generated training dataset since there were more than 453,900,000 million images. This dataset is referred to as pubchem_3.

RanDepict version 1.1.4 has been used to generate 127,500,000 hand-drawn-like synthetic structure depictions with an image size of 512 × 512 pixels using the pubchem_3 dataset. The augmentation functionalities that enable the generation of a hand-drawn-like style that has been implemented in RanDepict for this purpose are based on ChemPIX's implementation of hand-drawn-like hydrocarbon chemical depictions[58].

All training datasets were saved as TFRecord files to enable the training on TPU cloud instances using Tensorflow. Due to the large number of data points used in our training datasets (>400,000,000), the training dataset generation is a time-consuming process. Consequently, the SMILES datasets were divided into 100 chunks of equal length and used as input for the RanDepict toolkit, which was instantiated with different seeds to produce different sets of depiction features in each instance. To create TFRecord files directly from SMILES input, a custom Python script was used which is available in the RanDepict repository. The 100 SMILES list chunks per training dataset were processed on an in-house cluster using the workload manager Slurm. In each instance, 20 threads were used on virtual machines with 36 processor cores (2x Intel Xeon Gold 6140 18 Core 2,3 GHz) and 192 GB of RAM. Generating the datasets with an image size of 512 × 512 pixels took almost two weeks.

**Model selection.** DECIMER Image Transformer is based on an encoder-decoder architecture. A convolutional neural network (CNN) encoder generates feature vectors from 2D images which are then decoded by a transformer model[59] to yield a SMILES representation of the depicted molecule. The CNN encoder architecture used for DECIMER Image Transformer is EfficientNet-V2[60]. Specifically, the EfficientNet-V2-M CNN model was used for our work without any further modifications in order to accommodate the 512 × 512 image input size. In summary, EfficientNet-V2 incorporates the use of MBConv[61], including fused-MBConv[62], within its convolutional layers. In these MBConv layers, a smaller expansion ratio is employed to minimise memory access overhead. Additionally, a kernel size of 3 × 3 is utilised, accompanied by an increased number of layers to offset the reduced receptive field. In total, the model utilises 52 million parameters. For further information, we would like to refer the reader to the original publication.

The transformer model employed in this study is based on the model introduced in the 2017 publication titled "Attention is All You Need." It consists of four encoder blocks and four decoder blocks and incorporates eight parallel attention heads. The attention mechanism employed in this model has a dimension size of 512, while the feed-forward networks within the model have a dimension size of 2048. In total, the model utilises 59 million parameters.

**Training.** All of the DECIMER Image Transformer models were trained on TPUs available on the Google Cloud Platform (GCP). For training models, GCP offers a variety of TPUs. In this work, TPUs were selected for training models primarily due to their faster training speed,

scalability, and availability on the Google Cloud Platform. To enable the training on TPU devices, all datasets were saved as TFRecord files.

The training of models that were trained on the datasets pubchem_1 and pubchem_2 was run using a TPU V3-32 pod slice. The TPU V3-32 pod slice consists of four devices, which equals 32 nodes in total. This results in a fourfold increase in training speed compared to the previously used TPU V3-8 devices. The model trained using the pubchem_3 dataset was trained on a TPU V3-256 pod slice.

All models were trained using the Adam optimiser with a custom learning rate scheduler. Sparse categorical cross entropy was used as a loss metric. The dropout rate was set to 0.1 to avoid overfitting. When training models using the images with the size of $512 \times 512$ pixels, the per-node batch size was set to 48. Training scripts and models are written in Python 3 with Keras and Tensorflow 2.8.0.

**Computational considerations.** Training a model with the training dataset pubchem_1 on the TPU V3-8 device took nearly 3 days and 10 h on average per epoch. Training the same model using a TPU V3-32 pod slice took an average of one day and two hours. Thus, it was decided to train all models on TPU pod slices of V3-32 or higher to speed up the training process.

To train the models using the training dataset pubchem_3, the encoder had to be configured to accommodate the larger image size. Three EfficientNet-V2 encoder models were used to train and test the models trained using mages with the size of $512 \times 512$ pixels. These are EfficientNet-V2-B3, EfficientNet-V2-S, and EfficientNet-V2-M.

The training of the models with EfficientNet-V2-B3 per epoch took an average of 2 days and 3 h on a TPU pod slice V3-32. All training processes were moved to a TPU pod slice V3-256 to speed up training. Using EfficientNet-V2-B3, a model could be trained within 12 h and 30 min on average per epoch after changing the training device. For the model with EfficientNet-V2-S as the encoder, it took 15 h and 26 min on average to train each epoch, while for the model with EfficientNet-V2-M as the encoder, it required 1 day and 7 h.

**Test datasets.** In order to test the model trained on pubchem_3, the previously selected set of 250,000 molecules was used. Each of these molecules was depicted twice at $512 \times 512$ pixels, with and without augmentations. Moreover, to test the effectiveness of the model against images of chemical structures depicted with Markush structures, another dataset of 250,000 molecules, diversely selected using the MaxMin[51] algorithm and depicted using RanDepict, was used, where the images were shown both with and without augmentation at a resolution of $512 \times 512$ pixels.

**Evaluation of the test results.** The analysis of the test results was conducted using metrics generated with the CDK. All predicted SMILES strings for the test datasets were parsed using the CDK SMILES parser. Those that did not get parsed were labelled as invalid SMILES, whereas those that did get parsed were labelled as valid SMILES strings. Using the valid SMILES strings, accuracy and similarity were calculated by comparing each predicted SMILES string with the original SMILES string.

We initially generated InChI strings from the original and predicted molecules and compared them one to one in order to determine the accuracy of the model. For models trained using images with R-Group labels, obtaining InChI strings to calculate identical string matches is not possible. In order to overcome this problem, all of the original and predicted SMILES were parsed using the CDK SMILES parser, and an Isomeric CX SMILES was generated by combining CDK's Absolute and CXSMILES flavours. The SMILES string generated using this method then consists of a canonicalised SMILES with a '*' symbol where the R-Group should be present. At the end of each SMILES string, the R-Group labels that need to be inserted for the asterisks are listed. Using this particular SMILES variant, a one-to-one string comparison was performed, to determine the proportion of identical predictions.

Considering that the DECIMER Image Transformer model could potentially predict similar, but not identical molecules, it is important to also examine the similarity of the predicted molecules. Each predicted and original SMILES string pair was converted into CDK's iAtomContainer objects, and a Tanimoto similarity index was calculated based on PubChem fingerprints for each pair of original and predicted structures.

After all, the metrics have been calculated for each pair in the test dataset. The proportion of valid SMILES predictions, invalid SMILES predictions, the average accuracy, the average Tanimoto index and the proportion of Tanimoto 1.0 occurrences were calculated for each test dataset.

The BLEU (bilingual evaluation understudy) scores were calculated in addition to determining the accuracy of the predictions made by the model that predicts SMILES for images with Markush structures. This score evaluates how well a model can predict SMILES that are similar to the original molecule's SMILES.

**Evaluation of different R-group representations in SMILES.** Many Markush structures have more than one R-group attached to them. Therefore, the R-groups are commonly assigned indices, as in 'R$_1$' or 'R$_2$'. When creating SMILES strings with R-group representations, this leads to the introduction of tokens with multiple meanings. For example, the token '1' in the SMILES string 'c1ccccc1[R1]' can represent a ring opening or closure or an R-group index. To evaluate the influence of this potential problem, the performance of two models was compared.

In order to assess if these different possibilities of interpretation of the same tokens have an impact on the performance, two models were trained and tested. The first model was trained on images with R-group depictions and SMILES strings with R-group labels without any further modifications. The second model has trained on the same images, but the matched SMILES strings were modified to avoid tokens with multiple meanings. Every digit that occurs right after an R-group label is replaced by a character that does not have any function in the SMILES syntax. The following replacement characters were used:

$1 \rightarrow !, 2 \rightarrow \$, 3 \rightarrow \hat{\ }, 4 \rightarrow <, 5 \rightarrow >, 6 \rightarrow ?, 7 \rightarrow £, 8 \rightarrow ¢, 9 \rightarrow €, 0 \rightarrow §$

For example, this converts the SMILES string 'C[R5] N1C=NC2=C1C(=O)N(C(=O)N2C)C[R12]' into 'C[R>]N1C=NC2=C1C(=O) N(C(=O)N2C)C[R!\$]'.

The SMILES representations of Markush structures were downloaded from the SwinOCSR[24] repository and the images were generated using the CDK depiction generator with a resolution of $299 \times 299$ pixels. Character-based tokenisation was applied in both cases. Both models were trained on a set of 1 million structure depictions and tested on a set of 102,400 molecules from the whole dataset (selected using the MaxMin[51] algorithm), which were depicted as separate images; these molecules were not included in the training data.

For the evaluation, the original digits were re-inserted into the SMILES strings predicted by the second model. The SMILES strings were canonicalised and the Tanimoto similarity based on PubChem fingerprints was computed using the CDK. The performance evaluation was done based on the average Tanimoto similarity, the proportion of Tanimoto similarity values of 1.0, the proportion of exact string matches based on the canonical SMILES and the proportion of valid predicted SMILES representations of molecules (Fig. 7).

The model that was trained on the modified SMILES representation of Markush structures outperforms the model that was trained on the original SMILES representations. It yields a higher proportion of valid predicted SMILES strings (+3.4%), a higher proportion of Tanimoto 1.0 similarities (+2.2%) and a higher average Tanimoto similarity (+0.04), although the number of perfect predictions is slightly worse (−0.5%) for more details see Supplementary Table 3.

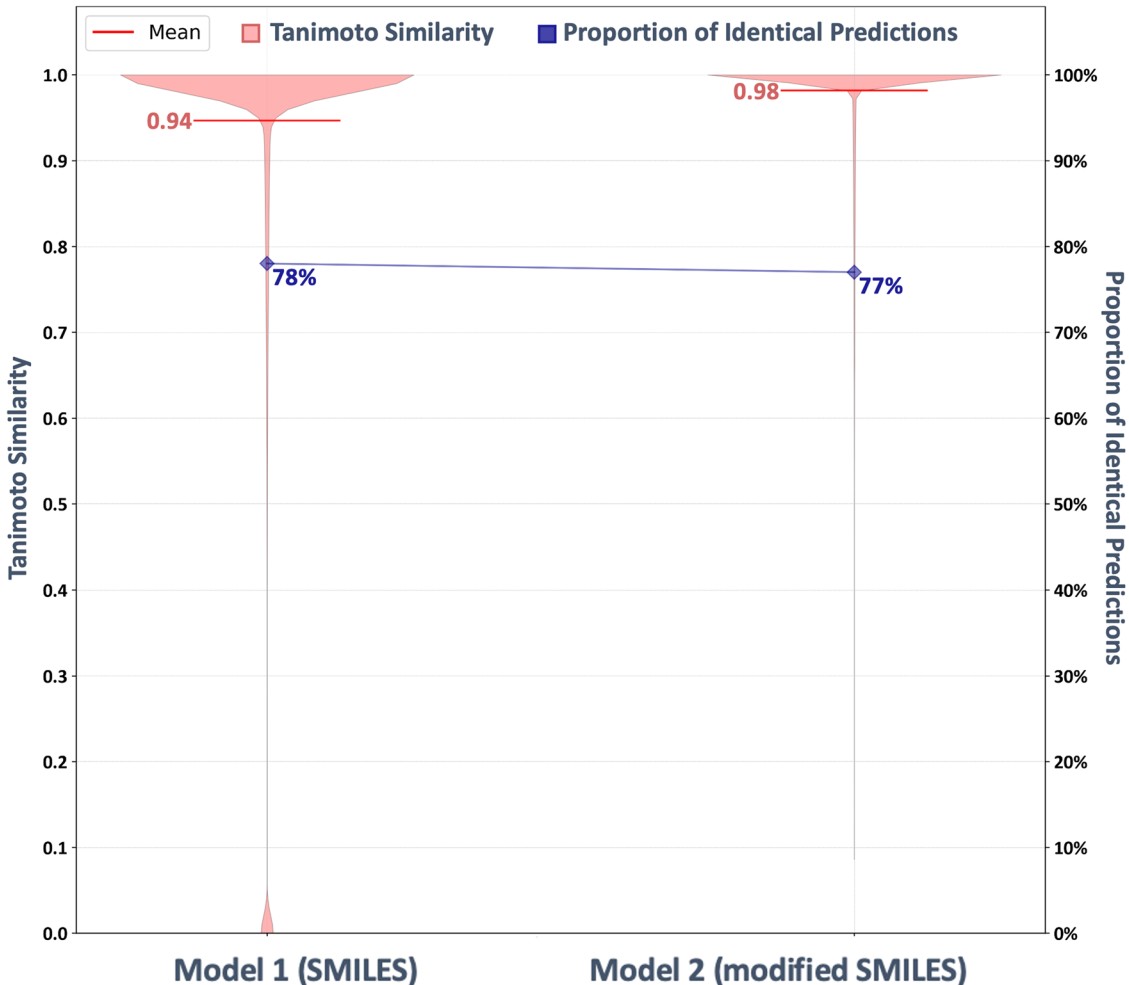

**Fig. 7 | Evaluation of the effect of the representation of R-group indices in the training data.** Test performance of a model trained on SMILES strings without further modifications (Model 1) and SMILES strings with replaced R-group indices (Model 2).

The results show that the double meaning of tokens in the training data (digits as part of the ring syntax or as an R-group index) leads to a lower performance of the model trained with it. Based on this finding, the modified SMILES representations were used for the training of all DECIMER Image Transformer models described in this publication.

**Improvements of the DECIMER Image Transformer.** The DECIMER Image Transformer has undergone significant advancements throughout its development, surpassing the capabilities of its initial iteration[18]. In the original version, a predefined set of rules was employed for dataset curation. However, in the current version, the application of such rules was omitted, and instead, molecules below a mass threshold of 1500 Da were included. Furthermore, the maximum output string length was increased. Previously restricted to a maximum of 66 characters for SELFIES strings, the current version supports SMILES strings with a maximum length of 150 characters. These enhancements have facilitated the incorporation of a more diverse range of molecules, enabling comprehensive coverage of an expanded chemical space beyond what was previously achievable.

In the original implementation, the SELFIES string representation was utilised. However, the current version has adopted SMILES as the preferred string representation. This decision was based on the benefit of a reduced token space offered by SMILES, which leads to better overall accuracy as explained in our previous work on molecular string representations[55].

A significant improvement has also been made regarding the depiction of molecules. In contrast to the initial implementation that solely relied on CDK-depicted[28] images with default depiction parameters, the current training data incorporates images depicted using RanDepict[37]. It features a diverse range of depiction styles and augmentations. Additionally, the canvas size has been expanded from 299 × 299 in the original version to 512 × 512 in order to accommodate larger and more complex molecular structures.

Regarding the improvements of the model, the initial implementation featured a pre-trained EfficientNet-V1 encoder with a Transformer as a decoder. DECIMER Image Transformer V1 used pre-trained weights for the image feature extraction with the encoder and only the Transformer was trained to decode the feature map. In contrast, in the current iteration, an EfficientNet-V2 CNN model has been integrated as an encoder in combination with the Transformer decoder, and both have been trained which resulted in a significant enhancement of the model's performance.

In terms of training, the advantages of EfficientNet-V2 have enabled the training of the models at a significantly faster pace, compared to models utilising EfficientNet-V1. To check the real-world performance of both the published models these were tested on the four OCSR benchmark datasets[63]. The benchmark results presented in Table 5 clearly demonstrate that the improvements in data generation techniques and model optimisation have resulted in the development of an enhanced model.

**Table 5 | Performance of DECIMER Image Transformer V1 and V2 on four benchmark datasets**

| | DECIMER V1 | | DECIMER V2 | |
|---|---|---|---|---|
| | $P_i$ | $T$ | $P_i$ | $T$ |
| JPO | 0.22% | 0.33 | 64% | 0.93 |
| CLEF | 0.30% | 0.37 | 72% | 0.96 |
| USPTO | 0.10% | 0.33 | 61% | 0.97 |
| UOB | 5.16% | 0.47 | 88% | 0.98 |

The performance is described as the proportion of occurrences of identical predictions $P_i$ and the average Tanimoto similarity $T$.

**Benchmark.** We determined the performance of the DECIMER Image Transformer and other available OCSR tools to assess their ability to be applied in a real-world use case to automate the mining of chemical structure depictions from the printed literature. A comprehensive benchmark of the DECIMER Image Transformer was conducted using all publicly available OCSR benchmark datasets and DECIMER test datasets.

The first four datasets were downloaded from Rajan et al. OCSR Review GitHub Page[63]. The other ones were generated or downloaded from the noted sources.

- USPTO: A set of 5719 images of chemical structures and the corresponding MOL files (US Patent Office) obtained from the OSRA online presence[64].

- UOB: The dataset of 5740 images and MOL files of chemical structures developed by the University of Birmingham, United Kingdom, and published alongside MolRec[65].

- CLEF: The Conference and Labs of the Evaluation Forum test set of 992 images and molfiles published in 2012[66].

- JPO: A subset (450 images and MOL files) of a dataset based on data from the Japanese Patent Office, obtained from the OSRA online presence[64]. Note that this dataset contains many labels (sometimes with Japanese characters) and irregular features, such as variations in the line thickness. Additionally, some images have poor quality and contain a lot of noise.

- RanDepict250k: A set of 250,000 chemical structure depictions generated with RanDepict (1.0.8) using RanDepict's depiction feature fingerprints[37] to ensure diverse depiction parameters. None of the depicted molecules is present in the DECIMER training data. The images here are all $299 \times 299$ pixels in size.

- RanDepict250k_augmented: A set of the same 250,000 images from the RanDepict250k dataset. Additional augmentations (examples: mild rotation, shearing, insertion of labels and reaction arrows around the structures, insertion of curved arrows in the structure) were added to the images using RanDepict. The images here are all $299 \times 299$ pixels in size.

- DECIMER hand-drawn[67]: A set of 5088 chemical structure depictions that were manually drawn by a group of 24 volunteers. The drawn molecules have been picked using the MaxMin[51] algorithm from all molecules in PubChem[50] so that the set represents a big part of the chemical space.

- Indigo: 50,000 images generated by Staker et al.[16] using Indigo[30] which were collected from the supplementary information. All images have a resolution of $224 \times 224$ pixels.

- USPTO_big: 50,000 images from the USPTO from Staker et al.[16] which were collected from the supplementary information. All images have a resolution of $224 \times 224$ pixels.

- Img2Mol test set: A set of 25,000 chemical structure depictions used by Clévert et al. for testing[15]. All images have a resolution of $224 \times 224$ pixels.

DECIMER Image Transformer was also benchmarked against a set of distorted datasets. These images were generated using the original OCSR benchmark datasets but with a slight shearing and rotation. The Img2Mol and DECIMER hand-drawn images datasets were not perturbed because they already contained a mixture of clean and perturbed images.

The following paragraphs describe the steps that were taken to run all the openly available OCSR tools. The compilation of OSRA with all of its dependencies is a complex task. To facilitate the usage, we have modified a version of docker-osra[68], a dockerised version of OSRA to update it to the newest version (at the time of publication: OSRA 2.1.3). The docker image of the version we used is available DockerHub[69]. To use it on our high-performance computing (HPC) cluster, the Docker image has been run with Singularity, an open-source containerisation application.

```
singularity run --bind /root_path/ docker://obrink/osra:2.1.3 sh /root_path/
scripts/run_osra_batch.sh /root_path/input_image/dir /root_path/
output_sdfile_path
```

The command above runs the script run_osra_batch.sh in the Docker image using Singularity. The script runs OSRA on every image in a given directory and saves the resolved structure as an SD file in a second given directory.

Content of run_osra_batch.sh:

```
#!/bin/bash
for image in $1/*.png;
do echo $image && osra -f sdf -w $2/${image##*/}.sdf $image;
done;
```

MolVec was downloaded as a jar file containing all dependencies[70]. It was used by running

```
java -jar /path/to/molvec-0.9.8-jar-with-dependencies.jar -dir /path/of/
input/image_dir/ -outDir /path/of/output/molfile/dir
```

Imago 2.0.0 was used via its command line utility with the compiled executable provided by the developer epam[71].

```
imago_console -dir /path/of/input/image_dir/
```

Img2Mol uses an encoder-decoder architecture. The original version of Img2Mol relies on an HTTP request of the encoded image to a server hosted by Bayer where the decoder is running. As the web server is only meant to be used for demonstration purposes, we contributed to Img2Mol to create a version that runs the decoder locally instead of sending HTTP requests to server[72]. This standalone version has been used to process all available benchmark datasets by running. The content of the script img2mol_batch_run.py is given in the supporting information (see Code Resource 1 in the supplementary information).

```
python img2mol_batch_run.py /input/path/ output/path png
```

As the original version of SwinOCSR did not include an inference script, we contributed to the open-source project to facilitate the usage of the model with the best performance according to the authors (focal loss model)[24]. After cloning the repository[73] and preparing the environment according to the instructions given there, it was used by running the following command in the directory that contains the scripts related to the above-mentioned model in the repository (SwinOCSR/model/Swin-transformer-focalloss/):

```
python run_ocsr_on_images.py --data-path /path/to/directory/with/images/
```

**Comparative performance evaluation of DECIMER Image Transformer and MolScribe after training on the same data.** Although MolScribe was trained on a significantly smaller dataset compared to DECIMER, its performance was similar. It is important to note that MolScribe and DECIMER employ different model architectures. MolScribe utilises an Image-to-Graph model, where atoms and nodes are predicted with coordinates and then reconstructed into a complete graph using a set of rules. DECIMER, on the other hand, follows a rule-free, purely data-driven approach in which the network is trained only on different images to learn to interpret chemical structures. To evaluate how well DECIMER performs after being trained on a smaller dataset, it was trained from scratch using the same dataset used to train MolScribe.

In MolScribe's training data pipeline, images are dynamically rendered during training using Indigo. Hence, we were not able to use the exact pipeline for the training dataset generation for DECIMER Image Transformer. The same molecules that are rendered during the training of Molscribe were depicted using Indigo's depiction functionalities in RanDepict and saved as a TFRecord dataset for the training of DECIMER Image Transformer. The USPTO training images were exactly the same in both cases. To ensure a proper evaluation of the different system architectures, they were both trained from scratch. Cross-validation was performed using MolScribe's ACS test dataset. Subsequently, the newly trained models underwent the aforementioned benchmarking process. However, to avoid potential overlap between the training datasets and images found in the USPTO and DECIMER benchmark datasets, these benchmark datasets were excluded from the study. The benchmark was done using six benchmark datasets comprising clean images, as well as four datasets containing distorted images.

Datasets with clean images:
- UOB: The dataset of 5740 images.
- CLEF: The dataset of 992 images.
- JPO: The dataset of 450 images.
- DECIMER hand-drawn: The dataset of 5088 images.
- Indigo: The dataset of 50,000 images.
- Img2Mol test set: The dataset of 25,000 images.

Dataset with distorted images (regenerated from the clean images):
- UOB: The dataset of 5740 images.
- CLEF: The dataset of 992 images.
- JPO: The dataset of 450 images.
- Indigo: The dataset of 50,000 images.

The results presented in Tables 6 and 7 clearly show that MolScribe outperforms DECIMER Image Transformer on all datasets when both models are trained on the same training data. The efficiency of MolScribe can be attributed to its model architecture and the post-processing workflow built into the MolScribe toolkit. The fact that DECIMER relies solely on a diverse set of training data without any predefined rules to gain its ability to interpret a chemical structure depiction means that its principal potential can not be realised with small amounts of training data, but requires large and diverse datasets.

## DECIMER Image Classifier

**Generation of chemical structure depictions.** Chemical structures were depicted as PNG images using the open-source toolkit RanDepict[37]. Five different chemical structure depictions were generated for each entry in the ChEMBL30[74] database (2,157,379 compounds) and the COCONUT database[75] (407,270 natural products). Once the chemical structure depictions were generated, the number of images without chemical structures was determined (6,814,929). In the

**Table 6 | Performance of MolScribe and DECIMER Image-Transformer on six clean benchmark datasets**

|  | MolScribe | | DECIMER | |
|---|---|---|---|---|
|  | $P_i$ | $T$ | $P_i$ | $T$ |
| JPO | 54.22% | 0.94 | 50.00% | 0.85 |
| CLEF | 74.34% | 0.96 | 63.54% | 0.89 |
| UOB | 87.67% | 0.99 | 69.97% | 0.95 |
| Hand_Drawn | 12.70% | 0.63 | 6.66% | 0.47 |
| Img2Mol_Test | 53.60% | 0.94 | 15.63% | 0.55 |
| Ingido | 39.41% | 0.98 | 12.24% | 0.62 |
| Average | 53.66% | 0.91 | 36.34% | 0.72 |

The performance is described as the proportion of occurrences of identical predictions $P_i$ and the average Tanimoto similarity **T**.

**Table 7 | Performance of MolScribe and DECIMER Image-Transformer on four distorted benchmark datasets**

|  | MolScribe | | DECIMER | |
|---|---|---|---|---|
|  | $P_i$ | $T$ | $P_i$ | $T$ |
| JPO_dist | 56.44% | 0.94 | 40.00% | 0.79 |
| CLEF_dist | 73.84% | 0.96 | 58.59% | 0.89 |
| UOB_dist | 87.25% | 0.99 | 55.42% | 0.91 |
| Ingido_dist | 36.12% | 0.97 | 2.66% | 0.40 |
| Average | 63.41% | 0.96 | 39.17% | 0.75 |

The performance is described as the proportion of occurrences of identical predictions $P_i$ and the average Tanimoto similarity **T**.

next step, the same number of images with chemical structures was randomly selected. Following the selection of images with chemical structures, the dataset was randomly divided into training, validation, and test sets based on the 80:16:4 ratio. The result was a training dataset containing 5,452,557 structure depictions, a validation dataset containing 1,089,899 depictions, and a test dataset containing 272,473 depictions.

**Generation and assembly of images without chemical structure depictions.** Using the matplotlib package in Python, 404,597 images of random graphs were generated with various options concerning plotting style, background, and text size. Additionally, we selected datasets containing images that could be mistaken for chemical structures, that could be easily presented in scientific papers or other diverse datasets (see Supplementary Table 4). In total, 6,410,332 images were retrieved from the public domain; a complete list of the datasets used can be found in Supplementary Information Table 4. In the same manner as the chemical images, the images with non-chemical data were randomly divided into training, validation, and test sets following an 80:16:4 ratio.

**Model architecture.** EfficientNet is a convolutional neural network architecture that has gained significant attention and acclaim for its efficiency and performance in image classification tasks. First introduced by Mingxing Tan and Quoc V. Le in 2019, EfficientNet[76] aims to achieve cutting-edge accuracy while preserving a compact model size and computational efficiency. Its notable innovation lies in its holistic approach to scaling the model uniformly in terms of depth, width, and resolution. Specifically, for the DECIMER Image classifier, we utilise the mobile-size baseline network, known as EfficientNet-B0. This network incorporates a multi-objective neural architecture that optimises both accuracy and FLOPS (Floating Point Operations Per Second). The core of this architecture revolves around the mobile inverted bottleneck

MBConv[61] block, which is commonly referred to as the inverted residual block and is further enhanced with an additional SE (Squeeze and Excitation) block[77]. The model hyperparameters include width and depth coefficients of 1.0, an image resolution of 224 × 224 pixels, and a dropout rate of 0.2%.

**Preparation and training.** DECIMER Image Classifier is based on the EfficientNet-V1-B0 model and was fine-tuned using 10,905,114 images, validated on 2,179,798 images, and tested on 544,946 images. The images were split randomly. With a batch size of 650 and five augmentations (vertical and horizontal flips, rotations, contrasts, and zooms), the whole training and validation process took about 52 hours and 15 minutes using a Tesla V100-PCI-E-32GB GPU.

**Performance evaluation.** The performance of the DECIMER Image Classifier was determined by evaluating its predictions on the test dataset. Initially, the Area Under the Curve (AUC) which measures the probability of correctly identifying instances more frequently than at random was calculated. Based on a value range from 0 to 1, 1 indicates an accurate classification, and 0.5 indicates total randomness. The DECIMER model's AUC for the test set is 0.99 (Supplementary Information Fig. 2).

The AUC allows the calculation of the highest distance between the curve and the random prediction. This is referred to as the Youden index (J)[78,79] and it reflects the model's threshold that achieves the best separation between the classes (chemical structure or no chemical structure).

Having established the most appropriate classifier threshold, other performance metrics using the confusion matrix can be computed, which include True Positive (TP), True Negative (TN), False Positive (FP), and False Negative (FN) values:

- sensitivity $= \frac{TP}{TP + FN}$; known as the true positive rate; the higher the score, the higher the proportion of TP in the set of positive predictions
- specificity $= \frac{TN}{TN + FP}$; known as the true negative rate; the higher score, the higher the proportion of TN in the set of negative predictions.
- Matthews Correlation Coefficient: MCC $= \frac{TP \times TN - FP \times FN}{\sqrt{(TP + FP)(TP + FN)(TN + FP)(TN + FN)}}$. The MCC is ranked between −1 and 1, where 1 represents a perfect classification while 0 represents a complete random sample.
- accuracy $= \frac{TP + TN}{TP + FN + TN + FP}$. This is the proportion of correct predictions for both classes.

**Test datasets.** The DECIMER Image Classifier was tested using four different datasets:

- ChEBI (Chemical Entities of Biological Interest)[80] database: This database was filtered to exclude structures found in ChEMBL and COCONUT databases, and five diverse depictions of each molecule were created using RanDepict, resulting in a total of 416,925 images.
- EM_Images (from Kaggle): This dataset contains 49,684 images of electron microscopy.
- PubLayNet[81]: This collection consists of 57,492 images illustrating figures from printed literature.
- JNP_real_world: A set of 8733 images that were automatically segmented from 1,000 publications from the Journal of Natural (JNP) products using DECIMER Segmentation. The segments were manually inspected by a human curator.

## DECIMER.ai

The DECIMER.ai web application has been developed using Laravel 8, a PHP-based web framework that follows the model-view-controller (MVC) design pattern. It runs as a three-container Docker application that can be deployed using docker-compose.

The three containers are responsible for running the nginx web server, communicating between the user interface and the processes running in the background and managing the deep-learning applications in the background.

When the app is launched, a user-defined number of socket server instances is started. Each of these socket servers listens to a different local port and waits to receive the location of an image to process. Multiple instances of each model type can be loaded. Working with multiple instances of these local socket servers allows fast parallel processing at the cost of constant memory usage for the preloaded models. This procedure was chosen to ensure a pleasant and fast user experience without the need to reload the models at every processing step.

Once the user uploads a PDF document, it is converted to multiple image files (one per page). The locations of these image files are then distributed over all available socket servers that run a preloaded model instance of DECIMER Segmentation. Once the chemical structures have been detected, the images are saved and their locations are sent back to the user interface where they are displayed. In parallel, the locations of the segments are sent to all available socket server instances that run preloaded instances of the models of DECIMER Image Classifier and DECIMER Image Transformer. The classifier instances receive the image path and return the values 'true' or 'false' based on whether the image is classified as a chemical structure depiction or not. The DECIMER Image Transformer instances receive an image path and return a resolved SMILES string. At this point, based on the SMILES strings, the corresponding molecules are displayed in the embedded Ketcher molecular editor[36] windows in the user interface and a warning is displayed if the image is not classified as a chemical structure depiction. Then, the user can download the segmented structure depictions, the corresponding MOL files and a file with the SMILES representations. If a single image is directly uploaded instead of a PDF, the same procedure of segmentation and subsequent OCSR processing is followed. If multiple images are uploaded, their locations are directly sent to the Image Transformer instances. If the user hits the button on the user interface, the resolved SMILES strings are sent to the STOUT[82] socket server instances, which return the corresponding resolved IUPAC names.

An instance of the DECIMER web application is available at https://decimer.ai. The complete source code is openly available on GitHub at https://github.com/OBrink/DECIMER.ai. The GitHub repository contains instructions on how to set up the web app locally and how to scale the memory requirements (as well as the parallel processing speed) by changing the number of socket servers with preloaded model instances. The instance of the web application running on https://decimer.ai is restricted to processing 10 pages and 20 structure depictions per uploaded document. The GitHub repository contains instructions on how to lift these restrictions when hosting a local version of the web application.

## Data availability

The datasets used for DECIMER Image Transformer were directly retrieved from PubChem: https://ftp.ncbi.nlm.nih.gov/pubchem/Compound/Extras/CID-SMILES.gz

The DECIMER Image Classifier training dataset is available on Zenodo[83].

The trained checkpoints of the models used in this study are available at:

-DECIMER Segmentation: Zenodo[84]

-DECIMER Classification: https://github.com/Iagea/DECIMER-Image-Classifier/tree/main/decimer_image_classifier/model

-DECIMER Image Transformer: Zenodo[85]

The benchmark datasets used in this study are available on Zenodo[86].

**Article** https://doi.org/10.1038/s41467-023-40782-0

The test and benchmark results generated in this study are provided in the results section of this publication and the Supplementary Information. Source data are provided with this paper.

## Code availability

The DECIMER Segmentation code is available on Zenodo and[84] at: https://github.com/Kohulan/DECIMER-Image-Segmentation

The DECIMER Image Classifier is available on Zenodo[83] and at: https://github.com/Iagea/DECIMER-Image-Classifier

The DECIMER Image Transformer is available on Zenodo[87] and at: https://github.com/Kohulan/DECIMER-Image_Transformer

The DECIMER.ai code is available on Zenodo[88] and at: https://github.com/OBrink/DECIMER.ai

The RanDepict code is available on Zenodo[89] and at: https://github.com/OBrink/RanDepict

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

## Acknowledgements
We are grateful for the company Google making free computing time on their TPU Research Cloud infrastructure available to us and K.R. acknowledges the research supported with Cloud TPUs from Google's TPU Research Cloud (TRC). We acknowledge Dr. Christoph Riplinger, FAccTs GmbH, Cologne, Germany, for the permission to use photos of the slides shown in Fig. 6. M.I.A. was supported by the Ministry of Education, Youth and Sports of the Czech Republic under project LM2023052. H.O.B., K.R. and C.S. acknowledge past funding for this work by the Carl-Zeiss-Foundation, and K.R. is currently funded by the German Research Foundation under project number: 239748522-SFB 1127 ChemBioSys (Project INF).

## Author contributions
K.R. & H.O.B. developed the software suite and performed the analysis. M.I.A. developed the DECIMER Image Classifier and implemented the hand-drawn-like augmentation features. K.R. and H.O.B. initiated, designed, tested, applied and validated the software features. C.S. and A.Z. conceived the project and supervised the work. All authors contributed to and approved the manuscript.

## Funding

## Competing interests
A.Z. is co-founder of GNWI-Gesellschaft für naturwissenschaftliche Informatik mbH, Dortmund, Germany. The remaining authors declare no financial and non-financial competing interests.
