## [Peer Review File · Nature Communications]

DECIMER.ai: An open platform for automated optical chemical structure identification, segmentation and recognition in scientific publicationsREVIEWER COMMENTS

Reviewer #1 (Remarks to the Author):

Recommendation: Minor Revision

Currently, most chemical information in the literature is still exclusively published in human-readable text and image formats, and manually converting images containing chemical structures into machine-readable representations is time-consuming and laborious. The manuscript presents an innovative open-source platform, DECIMER.ai, which can effectively identify, segment, and translate chemical structure depictions in scientific literature. Compared to other open-source OCSR software solutions, the platform employs a comprehensive workflow that integrates chemical structure image segmentation, classification, and translation.

However, the authors need to address the following concerns before the manuscript gets published:

1. In the experiment, the author considered various factors such as the Markush structure, R-Group label, etc. and achieved exceptional performance in all benchmark datasets. However, it would be more accurate if other models could train on the same dataset as DECIMER did.

2. It would be better if the authors could provide a more detailed description of DECIMER Image Classifier model and DECIMER Image Transformer model.

3. It would be helpful if the authors could provide a more detailed process of generating chemical structure images using Randepict in the data augmentation.

In addition, the following aspects can be considered by the authors for improvement in future:

1. It is worth noting that the platform restricts the number of pages in PDF files to less than 10 and the number of included chemical structure to less than 20. It is hoped that this restriction will be lifted in the future.

2. In addition, as the author said in the article, when the input structural image is blurry, the prediction accuracy is low, which can be improved in subsequent work.

Overall, the results of this study are highly relevant and significant to the field of cheminformatics and can greatly assist the research community.

Reviewer #2 (Remarks to the Author):

This work presents DECIMER.ai, an open source platform to recognize and translate chemical structure depictions from printed literature.

The authors have benchmarked their work extensively with other comparable methods to position their method.

As noted by the authors the main contribution is in combining image segmentation, classification and OCSR (decimer image transformer) in a user friendly web based tool.

I think both results and resulting application (code and datasets) are very useful contribution and have significance to the field of OCSR.

However, I would like the authors to make some clarifications. First of all, the authors cite their previous work [1] in the introduction but are not citing this work when introducing the Decimer Image Transformer which is their 'core component'. I find this a bit confusing as there are quite some similarities with this previous work [1] but also some differences. To better evaluate how this work is related and positioned in novelty with their previous work [1] I would like the authors to make clear what are the similarities and differences with the work [1]. I understand that the previous work [1] is also transformer based architecture but has SELFIES as output instead of SMILES. [1] is trained on a smaller dataset and the encoder is slightly different: EfficientNet instead of EfficientnetV2. Is this correct?

Concerning superatoms and functional groups:

The authors claim the application is capable of interpreting Markush structures as well as common functional groups and superatoms abbreviations. In the manuscript the authors focus on explaining how the application handles Markush structures with R-groups but I find no explanation (if not mistaken) on how the application handles the superatom abbreviations. For example are these split up in different tokens or as a separate token? Not clear for me now.

Concerning number of datapoints:

I find almost half billion of images as training data quite impressive. However some of the compared methods like MolScribe[2] use several orders of magnitude less data samples (200K range vs 400M range) as training data and also achieve quite good result. Also taken into account a recent paper[3] on OCSR, we see that methods based on object detection are more data efficient, I would suggest to add some discussion on this to the manuscript.

Some remarks on the structure of the manuscript:

I think some benchmark results in table form (now in supplementary material) would be better suited in the main text and maybe other material like the screenshots of the applications itself can move to supplementary materials.

Now the results are a bit scattered over the whole manuscript. There is some part in the beginning and some part in the method section and the most interesting results (table of performance on different datasets) are in the supplementary part. I would suggest replacing the barcharts, which plots the average performance over the benchmark datasets, with the table with the actual performance on the different datasets. These tables I find more useful for the reader to evaluate and compare the performance.

[1] Rajan, K., Zielesny, A. & Steinbeck, C. DECIMER 1.0: deep learning for chemical image recognition using transformers. *J Cheminform* 13, 61 (2021).

[2] MolScribe: Robust Molecular Structure Recognition with Image-to-Graph Generation
Yujie Qian, Jiang Guo, Zhengkai Tu, Zhening Li, Connor W. Coley, and Regina Barzilay
Journal of Chemical Information and Modeling (2023)

[3] Hormazabal, R., Park, C., Lee, S., Han, S., Jo, Y., Lee, J., Jo, A., Kim, S.H., Choo, J., Lee, M. and Lee, H., 2022. CEDe: A collection of expert-curated datasets with atom-level entity annotations for Optical Chemical Structure Recognition. *Advances in Neural Information Processing Systems*, 35, pp.27114-27126.

Dear reviewers,

We would like to thank you for your detailed evaluation of our manuscript and for the opportunity to amend and improve our submission. Herewith, we submit a revised manuscript where we have made changes according to your comments. We have addressed every comment in detail below and hope the revised manuscript meets your standards.

Kind regards,

Kohulan Rajan, Otto Brinkhaus, M. Isabel Agea, Achim Zielesny, and Christoph Steinbeck

REVIEWER COMMENTS

Reviewer #1 (Remarks to the Author):

Recommendation: Minor Revision

Currently, most chemical information in the literature is still exclusively published in human-readable text and image formats, and manually converting images containing chemical structures into machine-readable representations is time-consuming and laborious. The manuscript presents an innovative open-source platform, DECIMER.ai, which can effectively identify, segment, and translate chemical structure depictions in scientific literature. Compared to other open-source OCSR software solutions, the platform employs a comprehensive workflow that integrates chemical structure image segmentation, classification, and translation.

However, the authors need to address the following concerns before the manuscript gets published:

1. In the experiment, the author considered various factors such as the Markush structure, R-Group label, etc. and achieved exceptional performance in all benchmark datasets. However, it would be more accurate if other models could train on the same dataset as DECIMER did.

ANSWER: We agree with reviewer 1 that training on the same dataset might result in different results, and we can then select the best model architecture for OCSR applications. However, the purpose of this work and the benchmark is to compare the performance of the DECIMER Image Transformer model to other deployed tools as they have been published and not to assess the theoretical capabilities of different model architectures. In the future, it may be necessary to compare the model architectures to determine which application is most suitable for OCSR.

The training data used here has been generated using RanDepict and is suitable for training any OCSR tool that can be trained on pairs of structure depictions and string representations of chemical structures. In our benchmark, this is valid for Img2Mol and SwinOCSR (Molscribe needs additional atom coordinates which RanDepict currently cannot supply). However, training a deep neural network on a dataset of more than 400 million images requires a considerable amount of resources that we cannot invest in training the other available tools. If the developers of the other tools desire to improve them with the training data used for DECIMER, they can use our openly available training data generation pipeline. We do not consider the optimisation of other OCSR tools to be a part of the scope of our work.

Nevertheless, we have retrained the models of our strongest competitor Molscribe and DECIMER Image Transformer using Molscribe's training dataset to address this issue. This case study is presented in the Methods section of the revised manuscript (→ Comparative performance evaluation of DECIMER Image Transformer and MolScribe after training on the same data). We find that MolScribe performs better when trained on a comparably small dataset. This is likely due to DECIMER being an entirely data-driven approach

requiring more training data than MolScribe's graph assembly approach based on the predicted atoms and bonds. We have added a paragraph that discusses these results.

2. It would be better if the authors could provide a more detailed description of DECIMER Image Classifier model and DECIMER Image Transformer model.

ANSWER: We agree that there was only a small amount of information about the models used for the DECIMER Image Transformer and DECIMER Image Classifier. We have added more detailed descriptions of the used model architectures in the *Methods* section of the publication.

3. It would be helpful if the authors could provide a more detailed process of generating chemical structure images using Randepict in the data augmentation.

ANSWER: We agree that the general description of the generation of chemical structure depictions with Randepict should be explained in more detail. Consequently, we have extended the paragraph in the *Methods* section of the publication that explains it to provide a clearer overview to the reader.

In addition, the following aspects can be considered by the authors for improvement in future:

1. It is worth noting that the platform restricts the number of pages in PDF files to less than 10 and the number of included chemical structures to less than 20. It is hoped that this restriction will be lifted in the future.

ANSWER: The restriction to 10 pages and 20 structures per uploaded document is a measure that is in place to protect our web server from excessive usage. However, any user who wants to process larger documents using the DECIMER.ai web application can host their own local version of the application and remove the restrictions. The readme in the DECIMER.ai GitHub repository contains a link to a GitHub Wiki page with instructions on removing the restrictions.

To avoid confusion, we have mentioned the restrictions and the instructions for removing them in the *Methods* section of the revised publication.

2. As the author said in the article, when the input structural image is blurry, the prediction accuracy is low, which can be improved in subsequent work.

ANSWER: We are continuously working on improving our training data pipeline. The performance improvement on very blurry images could be achieved by implementing multiple resizing steps that lead to the occurrence of blurrier images in the training data. We have opened an issue in the Randepict repository (<https://github.com/OBrink/RanDepict/issues/37>) and will include more blurry images.

Overall, the results of this study are highly relevant and significant to the field of cheminformatics and can greatly assist the research community.

Reviewer #2 (Remarks to the Author):

This work presents DECIMER.ai, an open-source platform to recognize and translate chemical structure depictions from printed literature.

The authors have benchmarked their work extensively with other comparable methods to position their method.

As noted by the authors the main contribution is in combining image segmentation, classification and OCSR (decimer image transformer) in a user-friendly web-based tool.

I think both results and resulting application (code and datasets) are very useful contributions and have significance to the field of OCSR.

1. However, I would like the authors to make some clarifications. First of all, the authors cite their previous work [1] in the introduction but are not citing this work when introducing the Decimer Image Transformer which is their 'core component'. I find this a bit confusing as there are quite some similarities with this previous work [1] but also some differences. To better evaluate how this work is related and positioned in novelty with their previous work [1] I would like the authors to clarify the similarities and differences with the work [1]. I understand that the previous work [1] is also transformer-based architecture but has SELFIES as output instead of SMILES. [1] is trained on a smaller dataset and the encoder is slightly different: EfficientNet instead of EfficientNetV2. Is this correct?

ANSWER: A comparative summary of all changes and improvements would help the reader understand the system's development. As suggested, we cited the DECIMER Image Transformer 1.0 publication and added an extensive paragraph discussing the new model's improvements with a comparative benchmark study in the Methods section.

Concerning superatoms and functional groups:

2. The authors claim the application is capable of interpreting Markush structures as well as common functional groups and superatoms abbreviations. In the manuscript the authors focus on explaining how the application handles Markush structures with R-groups but I find no explanation (if not mistaken) on how the application handles the superatom abbreviations. For example are these split up in different tokens or as a separate token? Not clear for me now.

ANSWER: In the training data generation pipeline, the functionalities of the CDK, RDKit and Indigo are used to replace given structural elements with text labels that represent them in the structure depictions. This way, the token space is not affected at all. For example, in some images in the training data, a phenyl group is represented as a fully drawn substructure, in others, it is represented as a text label ("Ph"), but in both cases, the part of the SMILES string that describes the phenyl group would be something like "C1=CC=CC=C1". We agree that this should be explained better and we have added a statement about it in the *Methods* section (→ Generation of chemical structure depictions).

Concerning number of datapoints:

3. I find almost half billion of images as training data quite impressive. However some of the compared methods like MolScribe[2] use several orders of magnitude less data samples (200K range vs 400M range) as training data and also achieve quite good result. Also taken into account a recent paper[3] on OCSR, we see that methods based on object detection are more data efficient, I would suggest to add some discussion on this to the manuscript.

ANSWER: The MolScribe model has been trained on 1 million synthetically generated images and 680,000 images collected from the US Patent and Trademark Office (USPTO). We agree that the results presented by the MolScribe developers are impressive, considering the small training dataset. The same is valid for the data-efficient object detection-based OCSR system presented in the publication by Hormazabal et al. [3]. It is very difficult to make clear statements about the capabilities of the different model architectures, as they have all been trained on different training datasets.

Nevertheless, we have retrained the models of our strongest competitor MolScribe and DECIMER Image Transformer using MolScribe's training dataset to address this issue. This case study is presented in the Methods section of the revised manuscript (→ Comparative performance evaluation of DECIMER Image Transformer and MolScribe after training on the same data). Here, the MolScribe performs better when

trained on a comparably small dataset. This is likely due to DECIMER being an entirely data-driven approach requiring more training data than MolScribe's rule-based graph assembly approach based on the predicted atoms and bonds. We have added a paragraph that discusses these results and the above-described matter.

Some remarks on the structure of the manuscript:

4. I think some benchmark results in table form (now in supplementary material) would be better suited in the main text and maybe other material like the screenshots of the applications itself can move to supplementary materials.

ANSWER: We agree that the detailed information in the benchmark result tables should be included in the main text. Consequently, they have been moved to the results section (Table 1 and 2 in the revised manuscript) right after Figure 3 which illustrates the results in a compressed format. We prefer to keep the screenshot of the presentation of the results in the web application as an illustration for the reader.

5. Now the results are a bit scattered over the whole manuscript. There is some part in the beginning and some part in the method section and the most interesting results (table of performance on different datasets) are in the supplementary part. I would suggest replacing the barcharts, which plots the average performance over the benchmark datasets, with the table with the actual performance on the different datasets. These tables I find more useful for the reader to evaluate and compare the performance.

ANSWER: We have moved the tables with the benchmark results into the *Results* section (Tables 1 and 2 in the revised manuscript). We would prefer to keep the bar chart (Figure 3 in the revised manuscript) as well as it provides an overview of the results. If we only include the tables, the reader may find it difficult to identify trends in the data. Hence, the figure with the bar charts contributes to the clear and concise presentation of the results.

[1] Rajan, K., Zielesny, A. & Steinbeck, C. DECIMER 1.0: deep learning for chemical image recognition using transformers. *J Cheminform* 13, 61 (2021).

[2] MolScribe: Robust Molecular Structure Recognition with Image-to-Graph Generation
Yujie Qian, Jiang Guo, Zhengkai Tu, Zhening Li, Connor W. Coley, and Regina Barzilay
Journal of Chemical Information and Modeling (2023)

[3] Hormazabal, R., Park, C., Lee, S., Han, S., Jo, Y., Lee, J., Jo, A., Kim, S.H., Choo, J., Lee, M. and Lee, H., 2022. CEDe: A collection of expert-curated datasets with atom-level entity annotations for Optical Chemical Structure Recognition. *Advances in Neural Information Processing Systems*, 35, pp.27114-27126.

REVIEWERS' COMMENTS

Reviewer #1 (Remarks to the Author):

Recommendation: Accept

Thanks for the revised version. The authors have replied the main questions raised by the reviewers and provided a thorough and well-written response letter. And the revisions the authors made have greatly improved the quality of the manuscript.

The additional detailed description of the DECIMER classifier and DECIMER Image Transformer supplemented by the authors will greatly assist readers in understanding the author's work. The authors also made several relevant supplements concerning the specific generation process of the chemical structure image, which will help readers understand the generation process of the image. In addition, the authors added a comparison with their previous work DECIMER V1. This comparison will aid the reader in understanding the specific differences between DECIMER.ai and the previous method, providing a clearer perspective on the novelty and advancements presented in the manuscript. The authors also added a comparison with MolScribe. DECIMER.ai adopts the image-to-sequence method, while MolScribe uses the image-to-graph method. The comparison of the two methods is helpful for the reader to understand the respective advantages of the current cutting-edge method.

It is believed that both the work done by the author and the published data will serve as a valuable contribution to the field of OCSR.

Reviewer #2 (Remarks to the Author):

I would like to thank the authors for addressing all my concerns.

Response to the reviewers

Reviewers comments:

Reviewer #1 (Remarks to the Author):

Recommendation: Accept

Thanks for the revised version. The authors have replied to the main questions raised by the reviewers and provided a thorough and well-written response letter. And the revisions the authors made have greatly improved the quality of the manuscript.

The additional detailed description of the DECIMER classifier and DECIMER Image Transformer supplemented by the authors will greatly assist readers in understanding the author's work. The authors also made several relevant supplements concerning the specific generation process of the chemical structure image, which will help readers understand the generation process of the image. In addition, the authors added a comparison with their previous work DECIMER V1. This comparison will aid the reader in understanding the specific differences between DECIMER.ai and the previous method, providing a clearer perspective on the novelty and advancements presented in the manuscript. The authors also added a comparison with MolScribe. DECIMER.ai adopts the image-to-sequence method, while MolScribe uses the image-to-graph method. The comparison of the two methods is helpful for the reader to understand the respective advantages of the current cutting-edge method.

It is believed that both the work done by the author and the published data will serve as a valuable contribution to the field of OCSR.

Reviewer #2 (Remarks to the Author):

I would like to thank the authors for addressing all my concerns.

Dear reviewers,

We would again like to thank you for your time and consideration. We believe that your constructive criticism has helped improve the quality of the revised manuscript greatly. We really appreciate the time that you have invested in our manuscript and would like to express our gratitude for your positive evaluation.

Kind regards,

Kohulan Rajan, Otto Brinkhaus, M. Isabel Agea, Achim Zielesny, and Christoph Steinbeck